# Carbapenem-resistant *Escherichia coli* exhibit diverse spatiotemporal epidemiological characteristics across the globe

Jiewen Huang[1,5], Chao Lv[2,5], Min Li[2], Tanvir Rahman [3], Yung-Fu Chang [4], Xiaokui Guo [2], Zhen Song[1], Yanan Zhao[1], Qingtian Li [1✉], Peihua Ni [1✉] & Yongzhang Zhu[2✉]

Carbapenem-resistant *Escherichia coli* (CREC) poses a severe global public health risk. This study reveals the worldwide geographic spreading patterns and spatiotemporal distribution characteristics of resistance genes in 7918 CREC isolates belonging to 497 sequence types (ST) and originating from 75 countries. In the last decade, there has been a transition in the prevailing STs from highly virulent ST131 and ST38 to higher antibiotic-resistant ST410 and ST167. The rise of multi-drug resistant strains of CREC carrying plasmids with extended-spectrum beta-lactamase (ESBL) resistance genes could be attributed to three important instances of host-switching events. The spread of CREC was associated with the changing trends in $bla_{NDM-5}$, $bla_{KPC-2}$, and $bla_{OXA-48}$, as well as the plasmids IncFI, IncFII, and IncI. There were intercontinental geographic transfers of major CREC strains. Various crucial transmission hubs and patterns have been identified for ST131 in the United Kingdom, Italy, the United States, and China, ST167 in India, France, Egypt, and the United States, and ST410 in Thailand, Israel, the United Kingdom, France, and the United States. This work is valuable in managing CREC infections and preventing CREC occurrence and transmission inside healthcare settings and among diverse hosts.

[1] Department of Laboratory Medicine, College of Health Science and Technology, Ruijin Hospital, Shanghai Jiao Tong University School of Medicine, Shanghai, China. [2] Department of Animal Health and Food Safety, School of Global Health, Chinese Center for Tropical Diseases Research, Shanghai Jiao Tong University School of Medicine, Shanghai, China. [3] Department of Microbiology and Hygiene, Faculty of Veterinary Science, Bangladesh Agricultural University, Mymensingh, Bangladesh. [4] Department of Population Medicine and Diagnostic Sciences, Cornell University College of Veterinary Medicine, Ithaca, NY, USA. [5] These authors contributed equally: Jiewen Huang, Chao Lv. ✉email: qingtianli@sjtu.edu.cn; nipeihua@126.com; yzhzhu@sjtu.edu.cn

Antimicrobial resistance is recognized as a major global public health challenge by the World Health Organization (WHO). Annually, more than 700,000 people succumb to infections caused by antimicrobial-resistant microbes. Experts predicted that by 2050, this number might reach 10 million with an estimated cost of $100 trillion[1,2].

Carbapenem-resistant *Escherichia coli* (CREC) are gram-negative bacteria resistant to carbapenems, a class of antibiotics considered the last resort for treating infections caused by multidrug-resistant bacteria[3,4]. In 2017, the WHO published a list of bacteria urgently needing new antibiotics. Among them, CREC was identified as one critical highest-priority organisms in the WHO's priority first list[5]. CREC can lead to severe infections, including intra-abdominal infections, pneumonia, urinary tract infections, and device-associated infections. The high levels of antibiotic resistance and frequent transmission between humans and animals make CRECs infection a major public health concern[6]. Moreover, CREC is also a major food safety concern[7].

CREC strains commonly possess resistance genes to multiple antibiotic classes, which can complicate the treatment of CREC infections due to additional antibiotic resistance genes[8]. The presence of virulence genes in CREC strains also affects the severity and outcome of the infection, complicating patient management and care. It was demonstrated that CREC had evolved into a complex resistant strain of bacteria, instead of a single and stable antibiotic-resistant strain[9]. Understanding the general characteristics of CREC strains in terms of antibiotic resistance and virulence is crucial for monitoring and managing their spread.

The occurrence of CREC has been documented in multiple regions across the globe, including North America[10], South America[11], Europe[12], Asia[13], Africa[14], and Australia[15]. However, the prevalence and spread of CREC exhibited notable regional variation. Several regions have been identified as hyperendemic, such as France[12], Italy[16], the United Kingdom[17] in Europe, China, and India[18] in Asia, as well as certain parts of South America where a higher number of cases have been reported. For example, one study observed the prevalence of ST167, ST405, ST410, ST361, and ST648 in 874 *E. coli* isolates carrying $bla_{NDM-5}$. These isolates were collected from 13 European Union or European Economic Area countries over 2012–2022[19]. Another study highlighted the growing concern of high-risk CREC isolates, specifically ST131, circulating within the intensive care units in China[20]. Furthermore, a growing body of research suggests compelling evidence of CREC transmission among humans, animals, and the environment[21–23]. However, there hasn't been currently comprehensive information on the global prevalence, transmission patterns, and central spread routes of various STs of CREC. The exponential development of genomics and sequencing technology has led to the expansion of diverse databases containing a wide range of genomic resources. Researchers have globally shifted their focus to genomic epidemiological studies of carbapenemase-producing *E. coli*, utilizing various methods and data[24,25].

The efficient dissemination of antibiotic resistance genes relies heavily on resistance plasmids. Horizontal transfer of resistance plasmids involves the transmission of antibiotic resistance genes between bacteria through conjugation, transduction, or transformation mechanisms[26]. Conjugative plasmids contain transfer genes, also called *tra* genes, that encode the required machinery for the transfer process. These genes facilitate the plasmid transfer between bacteria, thereby facilitating the dissemination of diverse traits, including antibiotic resistance, virulence factors, and metabolic capabilities[27]. Mobilizable plasmids lack the complete set of genes required for its own transfer between cells. But they can rely on the presence of other conjugative elements, such as conjugative plasmids or integrating conjugative elements, to facilitate their transfer. Both conjugative and mobilizable plasmids are designated as mobile plasmids in this study, signifying plasmids possess an inherent propensity to transfer. Mobile plasmid-carried genes exhibit greater transferability compared to chromosome-carried genes. Consequently, resistance genes located on mobile plasmids have varying capacities for horizontal transfer, leading to different risks of resistance transmission[28].

This study utilized 7918 CREC strains to investigate their genetic diversity, antibiotic resistance profiles, and virulence characteristics. Furthermore, this study provides important insights into carbapenem resistance genes (CRGs)' distribution and transmission patterns across various countries and continents. Likewise, we have observed prevailing patterns in the prevalence of different phylogroups and STs among CREC strains. A thorough analysis has been conducted to evaluate the presence of antibiotic genes on chromosomes and plasmids, especially predicted mobile plasmids. The study also performed simulations to analyze the global transmission patterns of three CREC STs (ST131, ST167, and ST410) across five continents. Finally, the study investigated the transmission of ST10 and ST38 strains across three to four continents. This approach aimed to provide valuable insights into the highly varied transmission patterns observed in CRECs.

## Results

We retrieved 11,091 genomic sequences of CRECs from the NCBI Pathogens database, all containing CRGs. We further analyzed 7918 CREC genomic sequences with detailed location and time information. Figure 1a displays the distribution of CRGs across different countries, focusing on those with a minimum of 100 CREC strains. We observed that $bla_{NDM}$ was more prevalent in countries in Asia, North America, Africa and Oceania (Fig. 1c and Supplementary Data 3). Phylogroup D exhibited a higher prevalence of $bla_{OXA-48-like}$, accounting for 41.42% (548/1323) of the total population. This gene was most prevalent in Europe, particularly in Spain (77.96%, 145/186) and France (72.13%, 458/635), as shown in Fig. 1a. Phylogroup C had the highest proportion of strains with two or more CRGs, accounting for 8.76%. This was followed by phylogroup B1, which accounted for 3.40%. Africa had the highest proportion of strains containing multiple CRGs among all continents, with a percentage of 14.04%. Among individual countries, India had the highest percentage at 12.14%, followed by Israel at 8.66%. Our study also found that the $bla_{KPC}$ gene was more prevalent among South American CRGs, accounting for approximately 67.59% (98/145), including 92.00% (23/25) in Colombia, 75.00% (54/72) in Argentina, and 65.38% (17/26) in Brazil. It is worth noting that the $bla_{KPC}$ gene also demonstrated a high prevalence in the United Kingdom, accounting for 33.20% (381/1147), followed by 26.80% (41/153) in Germany and 19.8% (276/1394) in the United States. In contrast, $bla_{VIM}$ and other CRGs were more widespread in phylogroups G and B2 than in other phylogroups, as depicted in Fig. 1c, d and Supplementary Data 3 and 4.

The distribution of CRGs varies among strains from diverse sample sources. $Bla_{OXA-48-like}$ was responsible for 30.87% (2208/7152) of CRECs isolated from human sources. The percentage was twice as high for animal-derived CRECs and six times as high for environmental-derived CRECs. $Bla_{NDM}$ and $bla_{KPC}$ were most prominent in environmental strains, comprising 73.96% (476/644) and 12.87% (83/644), respectively. $Bla_{NDM-5}$ and $bla_{KPC-2}$ emerged as predominant genotypes in these two CRG series. It is worth noting that the occurrence of $bla_{KPC}$ was more than seven times greater in environmental strains compared with human strains (Fig. 1b and Supplementary Data 2). Statistically significant disparities were noted in CRECs derived from animals and the environment during 2013-2017 and 2018-2023 ($p = 2.36E-16$). During 2018-2023, animal-derived CRECs constituted 84.82% of the overall animal CRECs. Similarly, environmental-derived

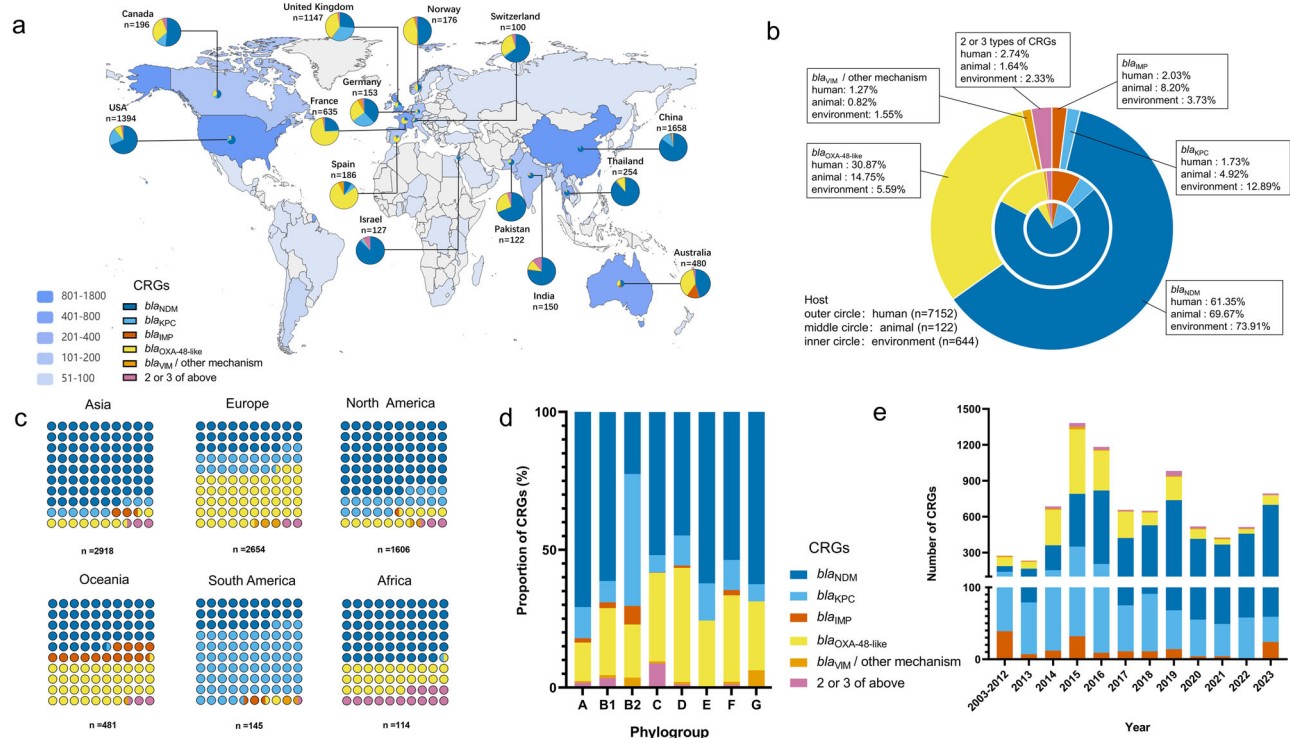

**Fig. 1 The global distribution of carbapenem-resistant genes (CRGs) in a dataset of 7918 CRECs. a** The global distribution of CREC and the proportion of different CRGs in countries where strains exceeding 100 have been reported. **b** A circular plot displaying the distribution of CRGs across various sources is represented as percentages. **c** The distribution of CRGs across the six continents. **d** The diversity of CRGs across different phylogroups of *E. coli*. **e** A histogram is presented below, illustrating the distribution of various CRGs expressed as percentages from 2003 to 2023.

CRECs accounted for 73.19% of the total environmental CRECs. On the other hand, no statistically significant disparity was observed in the overall number of human-derived CRECs across the periods. From 2003 to 2023, there was a considerable rise in the prevalence of $bla_{NDM-5}$ genes, whereas a decrease in $bla_{KPC-2}$ genes was observed. Most $bla_{OXA-48-like}$ genes notably peaked in 2015, followed by a subsequent reduction. Conversely, the prevalence of $bla_{IMP}$ genes experienced an overall decline between 2020 and 2022 before a resurgence in 2023 (Fig. 1e and Supplementary Data 5).

The 7918 CRECs have been discovered in over 75 countries, covering 497 unique STs. Notably, three STs (ST167, ST410, and ST131) were detected on all five continents, while six STs (ST38, ST405, ST361, ST648, ST69, and ST354) were found on four continents. Additionally, eight STs (ST10, ST617, ST156, ST448, ST46, ST1284, ST1193, and ST162) were identified on three continents. For example, ST167 strains predominated throughout Asia and North America. Spain had the most ST131 strains (42.30%), followed by China (19.05%), the US (18.85%), and the UK (17.45%). ST38 prevalence was 34.17% in Australia, higher than in Europe and Canada. Spain had the highest ST10 rate (34.90%), followed by Germany (21.57%) and France (16.39%). ST361 was most common in France (19.67%), Switzerland (15.52%), Thailand (11.40%), and India (10.73%) (Fig. 2a).

From 2013 to 2023, there was an upward trend in the prevalence of ST167, ST361, and ST648, while ST410 and ST38 showed a declining trend. Moreover, it is worth mentioning that the ST131 strain exhibited a decrease in prevalence from 2019 to 2020, followed by a subsequent increase from 2021 to 2023 (Fig. 2b, c and Supplementary Data 6, 7). About 36.63% (2900/7918) of all CRECs worldwide belong to phylogroup A, which mainly comprises ST167, ST361 and ST10. This is followed by phylogroup D, which accounts for 16.71% and primarily comprises ST38, ST405, and ST69. In the previous ten-year period,

there has been a notable rise in the relative representation of phylogroups A and B1. In contrast, the prevalence of phylogroups B2, C, D, and F has declined (Fig. 2b and Supplementary Data 6).

The fluctuation of STs in CRECs exhibits temporal variability. In 2013, the prevalence of ST131 strains exceeded that of all other STs. The ST410 shows a peak in 2016, followed by a subsequent decline observed between 2020 and 2023. In 2019, there was a notable rise in the prevalence of ST167, which has since maintained its majority, along with ST405, which also experienced an increase during the same period. The prevalence rates of the ST38 strains were found to be the highest in 2015 and 2017. Subsequently, there was a gradual decline until a resurgence was observed in 2023. Similarly, in 2023 a notable rise in the occurrence of ST10 was discovered, whereas ST361 exhibited the highest rate among the observed STs in 2022 (Fig. 3a and Supplementary Data 8).

This study demonstrated the prevalence of dominant STs of CRECs obtained from 14 countries. The predominant STs observed in China transitioned from ST131 in 2013 to ST167, ST156, and ST48 in 2023. This shift was accompanied by a decline in the prevalence of $bla_{KPC-2}$ and a rise in $bla_{NDM-5}$. From 2013 to 2023, there was a notable rise in the prevalence of $bla_{NDM-5}$-containing CRECs in China, with the percentage increasing from 6.78% (4/59) in 2013 to 69.41% (329/474) in 2023. Conversely, $bla_{KPC-2}$ decreased from 67.71% (37/59) in 2013 to 2.32% (11/474) in 2023. The ST131 strains have demonstrated a predominant ST in Spain in recent decades. The ST167 strains exhibited a high prevalence within the United States between 2019 and 2022, with almost all strains harboring the $bla_{NDM-5}$. The STs in Israel experienced a shift from ST410 from 2014-2015 to ST167 in 2018-2019. Similarly, in the United Kingdom, the primary ST shifted from ST216 in 2015 to ST38 in 2023. The prevalence of the $bla_{KPC-2}$ gene in ST216-CRECs

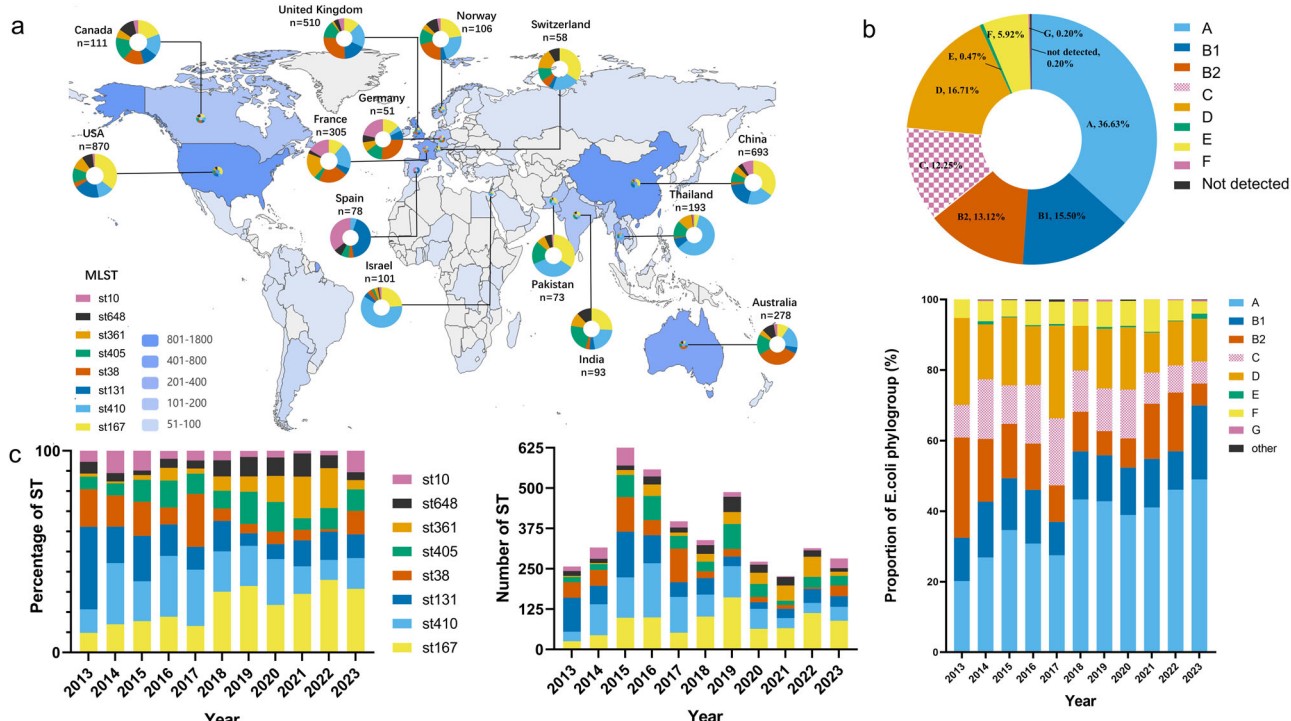

**Fig. 2 The prevailing Sequence Types (STs) and phylogroups among 7918 CRECs exhibit global spatial and temporal patterns. a** The map's color scheme represents the number of CRECs, while the pie charts for each country offer an overview of the global distribution of the most prevalent STs of CRECs. **b** Pie charts visually represent the proportional distribution of different phylogroups within 7918 CRECs. Histograms are graphical representations that depict the percentage distribution of various phylogroups of CRECs between 2013 and 2023. **c** The histogram displays the chronological distribution of the most prevalent STs from 2013 to 2023.

decreased from 99.3% to 77.8% (105/135) in ST38-CRECs, which now predominantly carry the $bla_{OXA-48}$ gene. Between 2013 and 2015, the predominant STs observed in France were ST38, ST10, and ST410, accompanied by a notable rise in the presence of $bla_{OXA-48}$. Nevertheless, during 2018-2022, ST361 emerged as the prevailing ST in France, as depicted in Fig. 3b. Our observations indicate that more than a decade subsequent to its emergence in 2012, $bla_{NDM-5}$ has evolved as the prevalent resistance gene with an enduring global augmentation.

Our results revealed an increase in the prevalence of 11 distinct categories of ARGs on plasmids from 2018 to 2023, compared to 2013 to 2017 (Fig. 4 and Supplementary Data 9). A significant rise in the majority of sulfonamide-resistant genes, with a jump from 65.99% to 80.47% ($p = 0.0207$). Furthermore, there was a substantial increase in the prevalence of extended-spectrum beta-lactamase (ESBL) resistance genes carried on plasmids, rising from 68.86% in 2013 to 88.79% in 2023. Moreover, the proportion of these genes carried on transferable plasmids also experienced an increase, from 7.89% to 15.12%. There was an observed increase in the proportion of quinolone genes harbored on plasmids, rising from 22.81% in 2013 to 37.81% in 2023. In addition, there was an increase in the prevalence of polymyxin resistance genes, as evidenced by the *mcr* gene on plasmids in 9% and 13.43% of CRECs in 2016 and 2023, respectively. Nevertheless, there was a decline in the prevalence of plasmid-borne CRGs from 86.64% during 2013–2017 to 79.85% during 2018–2023. Specifically, the proportion of CRGs on mobile plasmids decreased from 14.98% to 12.06%. Conversely, there was an increase in the occurrence of CRGs on chromosomes, rising from 31.92% to 45.71%. The prevalence of genes linked to lincomycin resistance was relatively low. However, in recent years, there has been a noticeable increase in prevalence of lincomycin resistance genes were observed.

Three prevalent STs, namely ST167 ($n = 913$), ST410 ($n = 862$), and ST131 ($n = 643$), from diverse geographical locations across the globe, were used to construct maximum likelihood phylogenetic trees. From 2014 to 2016, they exhibited the greatest prevalence of ST131, constituting 44.48% (286/643). The ST167 strains demonstrated the high prevalence rates in 2018, 2019, and 2022, accounting for 41.18% (376/913). From 2015 to 2017, the strain ST410 exhibited a high prevalence, accounting for 46.87% (404/862) (Fig. 5a–c). The CRGs in ST131 exhibited the highest occurrence of $bla_{KPC}$ genes, constituting 56.92% (366/643). The proportion of $bla_{OXA-48}$ and $bla_{NDM-1}$ was 15.71% (101/643). IncFII (pRSB107) was present in 41.68% of ST131, while IncFIB (pQil) and IncN accounted for 15.71% and 28.62%, respectively. Furthermore, it was observed that 56 strains exhibited the presence of $bla_{IMP}$ genes, suggesting that these genes were most prevalent among the three STs across all five continents (Fig. 5a). The ST410 strain demonstrated a remarkable prevalence of CRGs, with $bla_{NDM}$ accounting for 61.37% (529/862), followed by $bla_{OXA}$ at 40.02% (345/862). Additionally, it is worth noting that 77 strains belonging to ST410 were found to possess two or more CRGs, representing the largest number among the three STs analyzed (Fig. 5b). IncFIA and IncFIB (AP001918) demonstrated the highest in ST410 CRECs, with 89.68% and 88.98%, respectively. The ST167 strains exhibited the greatest percentage of CRGs containing $bla_{NDM}$, making up 94.19% (860 /913) (Fig. 5c). The prevalence of IncFIA and IncFIB in ST167 was lower compared to ST410-CREC, with proportions of 76.1% and 47.6%, respectively. Furthermore, the prevalence of IncI1 (Alpha), IncFII, and IncI (Gamma) in ST167 were 27.05%, 21.80%, and 13.47%, respectively.

ST167-CREC strains had 55.09% (503/913) mobile plasmids. ST131 was 46.03% (296/643), and ST410 was 30.97% (267/862). ST131-CREC had 37.64% (242/643) conjugative plasmids. ST167 was 30.01% (274/913), and ST410 was 25.41% (219/862). ST167

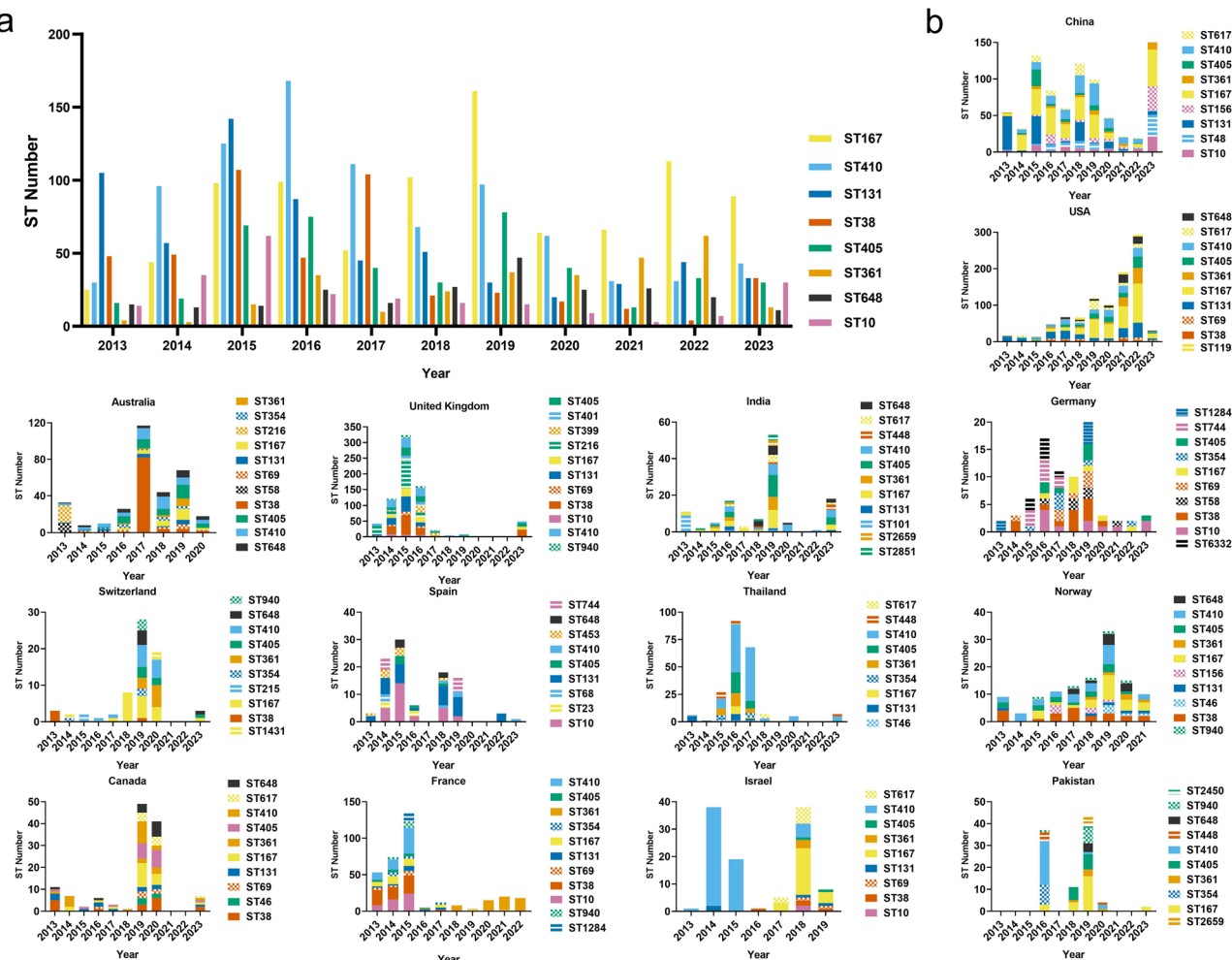

**Fig. 3 Frequency of the main Sequence Types (STs) from CRECs. a** The bar graph illustrates the frequency of STs displaying dominant CRECs from 2013 to 2023, specifically focusing on STs with strain numbers over 200. **b** The provided histogram shows the distribution of STs for the prevailing CRECs in countries with over 100 strains.

had the most mobilizable plasmids (33.30%, 304/913). ST131 (11.82%) and ST410 (8.24%) followed. ST131-CREC had the most transferrable antibiotic resistance genes (ARGs). It comprised 33.75% (217/643) of conjugative transferable CRGs and 73.31% of CRG-containing mobile plasmids. ST410-CREC had 12.88% (111/862), and ST167-CREC had 8.43% (77/913) mobile CRGs. The percentage of mobile plasmids harboring CRGs was at 41.57% (111/267) and 15.31% (77/503), respectively. Mobile ESBL genes were most common in ST167-CREC strains (16.65%), whereas ST131 had 8.24% and ST410 harbored 6.15% of such genes. ST167-CREC mobile plasmids had 61.54% ESBL genes, compared to 40.15% in ST131-CREC and 26.77% in ST410-CREC. ST167-CREC had the greatest rates of metastable aminoglycosides and sulfonamide resistance genes, 39.98% and 36.69%, respectively. These rates were more than three times ST131-CREC and ST410-CREC.

ST410, ST448, ST162, and ST156 are within the identical major branch, as depicted in Supplementary Fig. 1. There has been a notable shift from strains that exhibit higher virulence but lower levels of antibiotic resistance, such as ST405, ST131, ST38, and ST648, to strains that possess heightened resistance but display reduced virulence, exemplified by ST167, ST410, and ST361. The repertoire of genes associated with diminished virulence encompasses *chuS*, *espY1*, *kpsD*, *sat*, *shuA*, among others. Conversely, the augmented resistance genes contain $bla_{EC}$, $ble_{MBL}$, *mph*, *sul1*,

*tet(A)*, etc. The $bla_{NDM}$ gene was the most prevalent CRG, accounting for 64.68% of the 268 representative CREC strains analyzed. This was followed by the $bla_{OXA-48}$ gene, which accounted for 31.97%. Additionally, 14.87% harbored two or more CRGs. The aminoglycoside genes *aadA5*, *aph(6)-Id*, and *rmtB1* were undetected in the ST410 and ST361 strains. The gene $bla_{EC-15}$, which encodes for ß-lactamase, exhibited a predominant presence in ST167, ST410, and ST617, while the gene $bla_{EC}$ was primarily identified in ST361 and ST10. Thirteen isolates, comprising 4.85%, harbor *mcr* genes associated with polymyxin resistance, as depicted in Fig. 6a. About sixty-seven percent of CRECs had invasion-associated virulence factor *aslA*. Adhesion-related virulence factors were *fimA* (62.69%), *papB* (25%), *daaF* (8.21%), *draP* (9.33%), and *sat* (12.31%). ST131 and ST38 had higher *daaF* and *draP*. More than ninety percent of strains have siderophores-synthesizing gene *entD*. *FyuA* was found in 58.58% of 268 CRECs. The dominant ST167, ST10, ST361, and ST648 strains lacked the *fyuA* gene. Most ST131 strains lacked virulence genes *entD*, *espL4*, *espR1*, or *espX1*. Still, it had greater levels of invasion-associated *kpsD*, adhesion-like *papB*, secreted autotransporter *sat*, and enterotoxin-encoding *senB* than other STs.

Among the 268 CRECs, it is noteworthy that 36 strains are characterized by mobile plasmids that carry genes associated with ESBL production. The plasmid replicons IncFIA, IncFIB (AP001918), and IncX3 exhibited the highest frequency, with

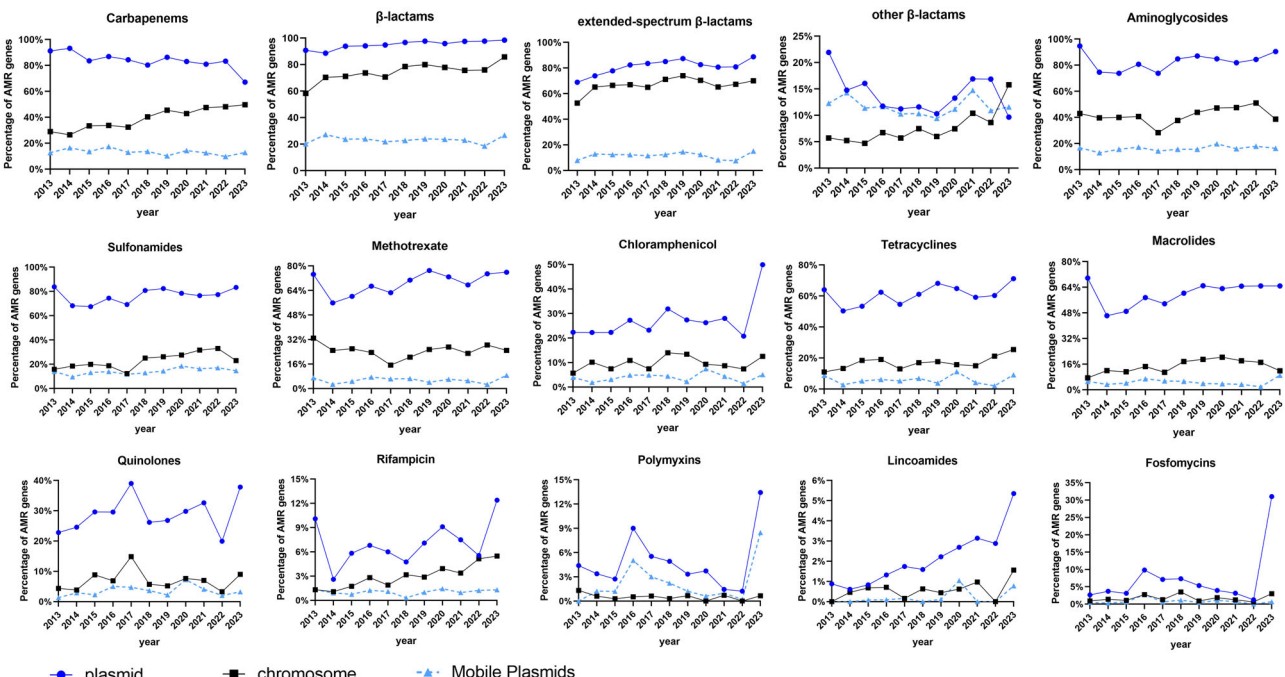

**Fig. 4 The distribution of antibiotic resistance genes (ARGs) among 13 antibiotic classes.** The dataset consists of 7918 CRECs that were collected between 2013 and 2023. The provided figure offers valuable insights regarding ARGs distribution across chromosomes and plasmids, especially mobile plasmids. The collection of mobile plasmids includes both conjugative and mobilizable plasmids. The illustrations are categorized into three groups: total β-lactams, ESBLs, and other β-lactams.

rates of 61.57%, 58.58%, and 32.5%. The antimicrobial resistance genes that showed the highest mobilization potential were those conferring β-lactamase resistance, with 55.34%. This was followed by genes conferring resistance to chloramphenicol, aminoglycosides, and tetracycline, with 34.95%, 28.16%, and 27.18%, respectively (Fig. 6b).

According to our results, the current CREC ST38 clones can be traced back to a zoonotic transmission event in about 1975. The CRECs associated with phylogroup A can be attributed to two host-switchings occurring between humans and the environment, specifically in about 1870 and 1931. It is hypothesized that these events may have contributed to the emergence of ancestral CRECs, specifically ST161, ST10, and ST361, identified from 2018 to 2023. The presence of contemporary CRECs that carry plasmids containing ESBL resistance genes can be attributed to three occurrences of host-switching between human populations, the environment, and animal species (Fig. 6c).

The transmission country hubs for ST131 strains were identified in the United Kingdom, Italy, the United States, and China across five continents (Fig. 7a). The ST167-CREC demonstrated central hubs in India, France, Egypt, and the United States. In contrast, the transmission of ST410-CREC occurred in Thailand, Israel, the United Kingdom, France, and the United States (Fig. 7b, c). The ST38 strains accounted for the highest number of CRECs across four continents, totaling 465 cases. These strains exhibited extensive transmission in France, the United Kingdom, Norway, and the United States. The number of transmission events in Australia escalated to 81, with the outbreak originating from Vietnam in 2017 (Supplementary Fig. 2). The ST10 CRECs exhibited the greatest abundance of CRECs across three continents, totaling 232, primarily distributed among China in Asia, the United States in North America, France, the United Kingdom, and Spain in Europe (Supplementary Fig. 3).

Utilizing a temporal evolutionary tree encompassing 643 ST131-CREC obtained from 38 countries across five continents

during 2005-2023, 96 transmission routes were identified. Forty-three of these transmission routes have been selectively displayed on the map (Fig. 7a). The intercontinental transmission routes of ST131-CREC involve multiple introductions within Europe, dissemination from Asia to North America through Africa, and transmission from North America to Asia via Europe. Many travels or trade-related transmission patterns have also been discovered in the United Kingdom and the United States. The study revealed importation events from Southeast Asia to Australia and South America and the subsequent transmission of the imported routes from Australia and South America to North America and Europe. The nations exhibiting the greatest prevalence of ST131 strains were the United States ($n = 164$, 2012-2023), China ($n = 128$, 2012–2023), the United Kingdom ($n = 89$, 2011–2018), Italy ($n = 43$, 2005-2020), and Japan ($n = 42$, 2011-2023). The primary transmission routes for the international dissemination of ST131-CRECs were as follows: there were 30 transmissions from the United Kingdom to the United States between 2014 and 2022, 23 transmissions from the United Kingdom to Italy during the period of 2015-2016, and 16 transmissions from China to Japan between 2015 and 2023 (Fig. 7a). The original strain of ST131, known as EC_BZ_1, was discovered to possess four distinct plasmid replicons, specifically IncFIA, IncFII, IncN, and IncX4. However, the analysis conducted on mob-typing did not yield any evidence of mobile plasmids. The EC_BZ_1 genotype homologous strain, specifically identified as ERS6266043, was observed in Argentina in 2010 and consisted of seven plasmid replicons, Col (MG828), Col156, ColRNAI, IncFIA, IncFII, and IncR. The mobile plasmid harbored three resistance genes, *sul1*, *qnrB2*, and *tet(A)*, and six virulence genes, *papB*, *sat*, *iucB*, *iucA*, *iucD*, *iucC*, and *papI*.

The transmission routes of the globally distributed ST410- and ST167-CRECs demonstrated variations compared to ST131, as illustrated in Fig. 7b, c. The ST167-CRECs were initially identified in Nepal and France and subsequently experienced swift

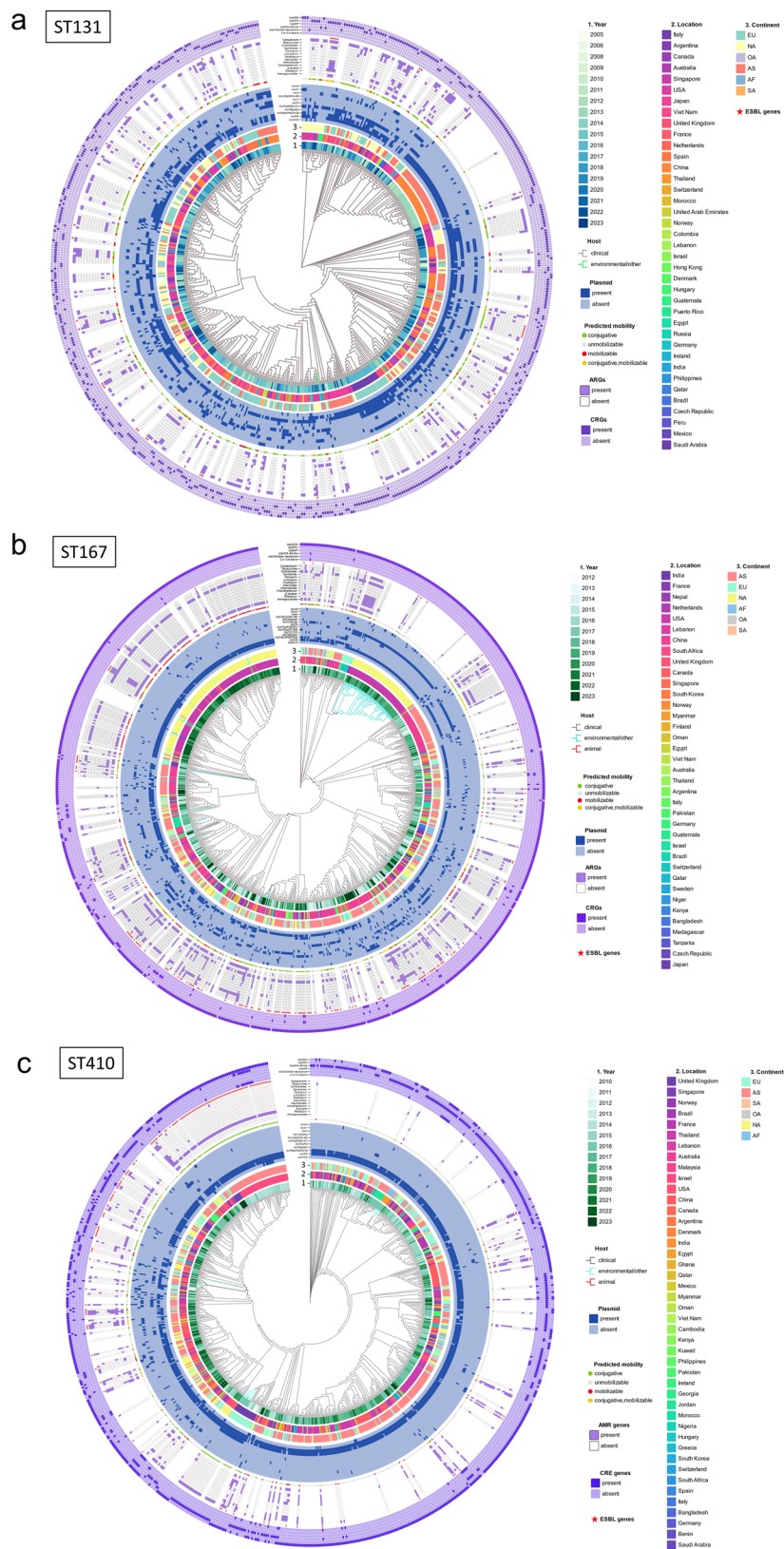

**Fig. 5 The circular diagram illustrates the phylogenetic tree of CRECs across five continents. a** Phylogenetic tree of 643 CRECs for ST131; **b** Phylogenetic tree of 913 CRECs for ST167; **c** Phylogenetic tree of 862 CRECs for ST410. The concentric circles in the diagram represent various information, including the year, location, continent, plasmid replicon, plasmid mobility prediction results, resistance genes associated with 13 antibiotic classes carried by mobile plasmids, presence of ESBL, and presence of CRGs carriage.

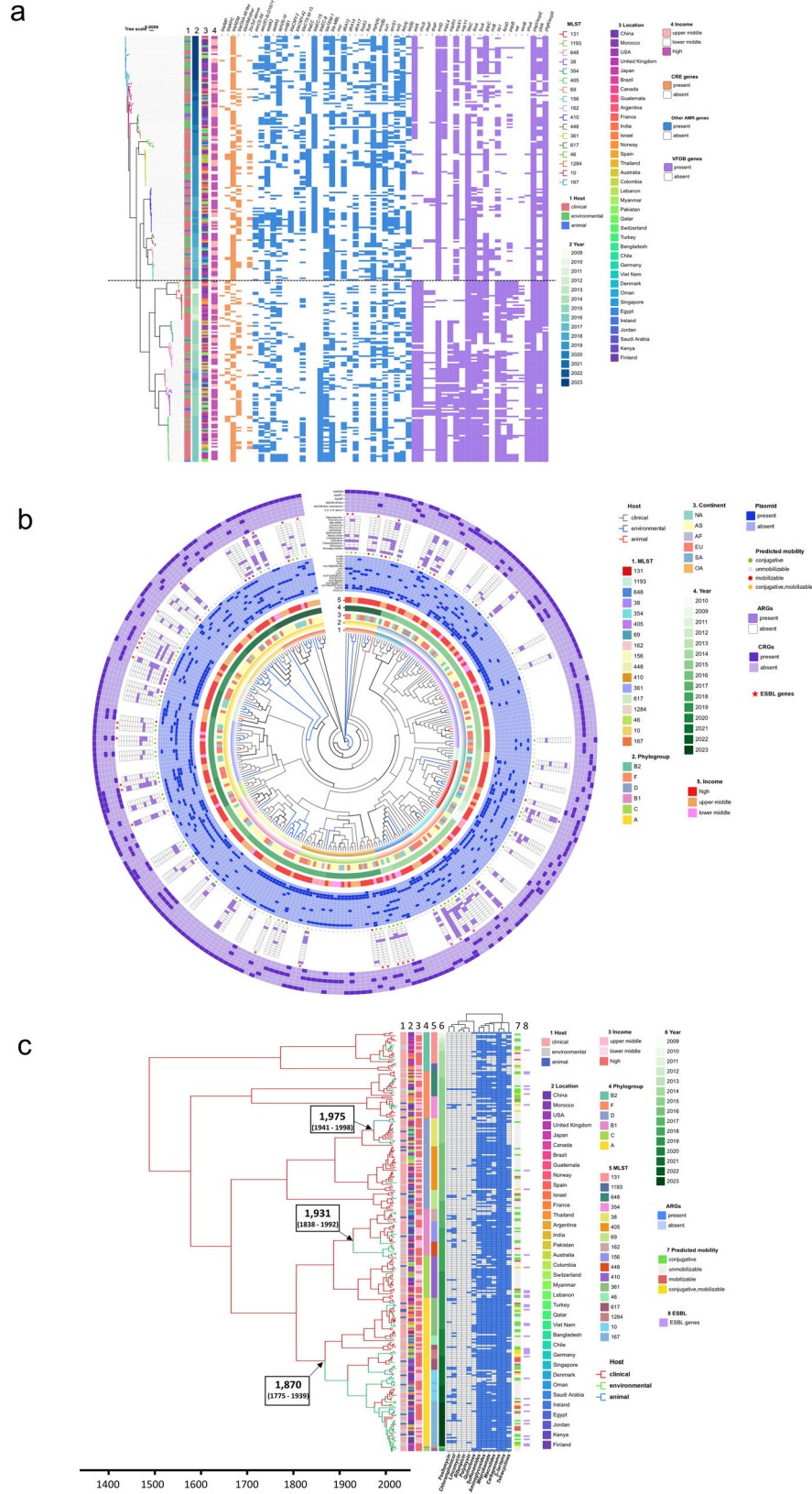

**Fig. 6 Phylogeny and global expansion of CRECs.** The phylogenetic tree contains 268 representative CRECs from each branch of the evolutionary tree of 17 mainstream ST strains spanning 3–5 continents. **a** Heat map dendrogram indicated STs, hosts, time, geographic location, country income, pathogenicity, and resistance profiles for 268 CRECs. **b** Staining of evolutionary tree branches shows the source of 268 CRECs. Concentric circles in the phylogenetic tree of 268 CRECs represent various information including the year, location, plasmid replicons and resistance genes. **c** 268 CREC time-scale tree with host ancestral reconstruction. The branches were colored by the host with the highest probability, and arrows pointed to the main evolutionary jump between humans and animals or the environment. Key node time ranges were confidence intervals.

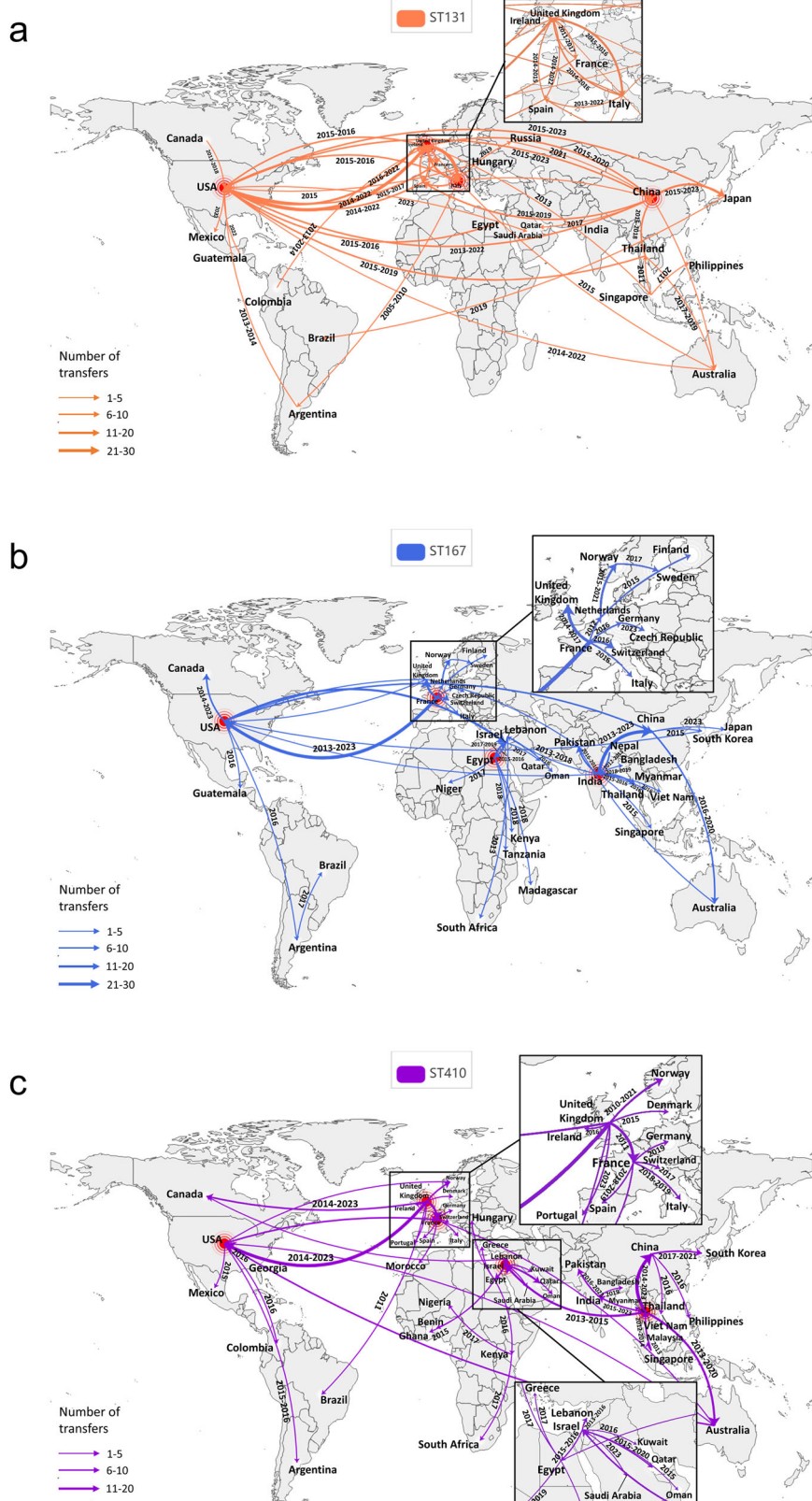

**Fig. 7 CREC transmission patterns of dominant STs on five continents. a** Dissemination patterns of ST131-CREC. **b** Dissemination patterns of ST167-CREC. **c** Dissemination patterns of ST410-CREC. The geographic spread of CREC has been inferred through ancestral state reconstruction of the timed phylogenetic tree. The thickness of the arrow lines is adjusted proportionally based on the number of transfers exchanged between countries. Time shows the transmission period of the geographical transfer of CRECs between each pair of countries. Intercontinental STs were classified as those with more than 10 CREC strains spreading across each continent. Countries indicated by red dots are those designated as having transmitted CRECs to five or more countries with a total of more than ten strains.

dissemination throughout Southeast Asia following their transmission from Nepal to India. Later, these ST167-CRECs extended their reach to China from 2013 to 2023. Similarly, there were observed in Australia from 2016 to 2020. The dissemination of French ST167-CREC across Europe was subsequently accompanied by ST167-CREC exhibiting a high similarity to European CRECs within the United States from 2013 to 2023. Between 2017 and 2019, there was an additional occurrence of the spread of ST167-CREC to Israel, originating from the United States. The proliferation of ST167-CREC clones in Africa was documented from 2015 to 2019. The ST410-CRECs, on the contrary, had their origins in the United Kingdom and subsequently disseminated across Europe, with further expansion into North America from 2014 to 2023. Thailand and Israel serve as central hubs for the dissemination of ST410 strains into the regions of Asia and Africa, respectively. Between 2013 and 2020, a homologous ST410 strain emerged in Australia from Thailand. Subsequently, from 2020 to 2022, the ST410-CREC was transmitted between Australia and the United States.

## Discussion

The present study analyzed the worldwide distribution of various CRGs, with 497 STs from 75 countries, elucidating the spatial and temporal patterns of prevalent STs and CRG prevalence across diverse evolutionary classifications of CREC. To the best of our knowledge, this should be the first systematic study of global distribution and transmission patterns of CREC and trends in dynamics at multiple national, international, and continental interfaces.

In our study, multiple categories of CRGs have exhibited distinct patterns over the last two decades. The $bla_{NDM}$ gene dominates CRECs, was particularly prevalent in Asia and North America, and has shown an unusual upward trajectory in global transmission. In particular, $bla_{NDM-5}$ has increased dramatically in transmission to pivotal countries such as the United States, France, and China. Conversely, the $bla_{KPC}$ gene, primarily associated with the phylogroup B2, showed a higher prevalence in South America and a declining trend. The $bla_{OXA-48}$ gene is frequently observed as a carbapenem resistance determinant in *E. coli*[29]. The $bla_{OXA-48-like}$ gene exhibited a high prevalence within phylogroup D and demonstrated a notably high occurrence in Europe. Analyses of the sample sources showed that the proportion of $bla_{OXA-48}$ in human CREC strains is greater than their respective proportions in animal and environmental CREC strains. It was observed that animal-derived CRECs exhibited a higher degree of prevalence during 2018-2023. Additionally, our study shows the prevalence of $bla_{IMP}$ genes was greater in animal sources than in human and environmental sources. It is widely acknowledged that bacteria interact with hosts, antibiotics, and environmental factors. As a result, bacteria from diverse sources possess various resistance and virulence genes, which lead to the exhibition of distinct phenotypic traits[30].

The findings of this study demonstrate a rise in the prevalence of major phylogroups A and B1 among CREC strains, whereas other phylogroups exhibited a decline. Several studies have reported that specific phylogroups of *E. coli* show a higher prevalence of drug resistance than other phylogroups[31,32]. Fortunately, there has been a discernible downward trajectory in the prevalence of phylogroup B2 over the past ten years[33]. This phylogroup B2 exhibits a propensity for heightened antibiotic resistance. The observed variations in the prevalence of different phylogroups across continents indicate that various factors, including human migration, commodity and animal trade, economic and health development, and disease management strategies, substantially influence the distribution and equilibrium of distinct phylogroups.

The dissemination of plasmids containing CRGs among human populations and in various environments, such as agricultural settings and other geographical locations, has already been documented[34]. Our findings revealed that the proliferation of sulfonamide resistance genes harbored on the plasmid was the most prominent, and 11 other resistance genes, such as ESBL, quinolone, and polymyxins, also exhibited some levels of propagation. Nevertheless, the prevalence of plasmid-encoded CRGs has declined significantly, indicating that implementing policies to restrict the use of carbapenem antibiotics in various countries following the WHO advisory has had some impact. The ancestral lineage of 268 representative CRECs was examined using a temporal evolutionary tree, revealing that the appearance of current multi-drug resistant strains and CRECs harboring plasmids with ESBL resistance genes can be ascribed to three pivotal instances of host-switching between humans, the environment, and animals. The events above were believed to have occurred in about 1975, 1870, and 1931. This phenomenon poses challenges to the management of CREC, necessitating a comprehensive approach to prevention and control encompassing multiple sectors and integrating perspectives from the "One Health" framework.

The study identified ST167, ST131, and ST140 as the most prevalent types of CREC. These types were found to have the highest strain numbers and have been widely distributed across five continents. Previous studies have demonstrated that these types possess the capacity to transport and disseminate genes associated with virulence or resistance, which has important implications in clinical settings[23,35,36]. The global dominant short-term for CREC has experienced continuous fluctuations over the past decade. The prevailing ST has gradually shifted from ST131 and ST38, with higher virulence but lower levels of antibiotic resistance, to ST410 and ST167, which exhibits higher resistance and slightly reduced virulence. ST131-CREC originated in Italy in 2005. As it spread, over one-third of the ST131-CRECs harbored conjugative plasmids, and it acquired mobile plasmids such as IncFII carrying resistance genes for sulfonamides, quinolones, and tetracyclines. In the United States, the strain showed an increase in the prevalence of $bla_{KPC-2}$ and $bla_{KPC-3}$, while the Chinese strain showed a decrease in $bla$NDM-5. In 2014, the UK ST131 strains showed $bla_{OXA-48}$. Our study demonstrated that over one-third of ST131-CRECs contained conjugative plasmids, including IncFII. Notably, this particular plasmid type plays a key role in facilitating the dissemination of carbapenem resistance. This plasmid can undergo gene rearrangement in CRECs from different sources carrying the $bla_{KPC-2}$ gene[37]. The ancestry of contemporary ST38 originated from a zoonotic transmission event in about 1975, with an increase in the $bla_{OXA-48}$ gene during its spread in Europe from 2014-2020. We also found an outbreak epidemic in Australia after being exported from Vietnam in 2017, generating many CRECs containing the $bla_{OXA-181}$ gene.

ST410-CREC first appeared in the United Kingdom in 2010. The prevalence of $bla_{OXA-181}$ increased steadily in Europe until 2016 when it gradually declined. Meanwhile, $bla_{NDM-5}$ increased significantly during transmission in Asia, reaching its peak in 2019. During the global spread of ST410-CREC, removable plasmids such as IncFIA and IncFIB carrying CRGs and ESBL resistance genes have emerged on several continents. In 2012, the ST167-CREC surfaced in France and Nepal, gradually emerging as the strains with the highest global transmissions. The presence of $bla_{OXA-48}$ witnessed a significant increase in its proliferation in Europe between 2014 and 2015, whereas $bla_{NDM-5}$ experienced a continuous escalation, eventually becoming the predominant CRG of ST167. Most of the ST167-CRECs observed in our study exhibited the presence of various mobile plasmids. Concurrently, there has been an increase in the levels of multidrug-resistant plasmid IncI during its dissemination. IncI-complex plasmids

carry a wide range of resistance genes, especially ESBL genes and CRGs. Forde's study found that the antibiotic resistance islands can transfer from the multidrug resistance IncF plasmid to the IncI plasmid[38]. ST167 CRECs exhibited a notable prevalence of mobile ESBL genes and a higher proportion of metastable aminoglycoside and sulfonamide resistance genes. The findings of our study indicate that ST167 showed a consistent increase over the past ten years. Since 2018, ST167, with high levels of resistance, has emerged as the most prevalent CRE strain spreading across five continents. Due to its extensive distribution, this dominant strain of carbapenem-resistant *E. coli* warrants great attention and scrutiny.

Investigating the extensive migration of pathogens over long distances using genomic and evolutionary indicators is a complex objective that has recently attracted considerable scholarly attention[39]. This study presented the credible evidence of the worldwide geographic dissemination and proliferation of CRECs by constructing temporal evolutionary trees and reconstruct host ancestry. The investigation unveiled the global dissemination patterns and transmission centers of major CREC STs across three to five continents. These transmission centers are critical for swift outbreak alerting and managing antibiotic resistant pathogen dissemination. According to our established criteria, we have identified the dissemination hubs of ST131 (United Kingdom, etc.), ST167 (India, etc.), and ST410 (Thailand, etc.).

The central hubs of the CREC are intricately linked to the transnational movement of individuals across various nations, encompassing economic endeavors and the tourism sector. One recent study examines the correlation between the transmission of distinct ST strains and the proliferation of diverse antibiotic resistance and virulence genes[40]. Hence, it is imperative to strengthen the monitoring and collaborative governance of highly resistant pathogens, such as CRECs, among nations and regions that share economic interdependencies and experience population mobility. To enhance the detection and control of outbreaks and resistance spread of these pathogens, it is crucial to focus on transmission hubs that migrate in multiple directions geographically. Possible relevant programs to consider may involve implementing strategies such as augmenting the frequency of entry and exit testing and concurrently implementing measures to reduce antibiotic utilization[41,42].

Previous studies on genomic data for CREC epidemiology have been limited to specific regions. However, in light of increasing global human mobility, travel, and trade, comprehensive analyses of all available sequences can yield valuable epidemiological insights on CREC at both local and international levels. Such studies can inform infection control measures by revealing CRG gene distribution, drug resistance gene mobility, and CREC geographical transfer patterns and hubs. As the data used for our analysis were sourced from the Pathogenic Infection Repository and reported by country, some geographic variability may have been unavoidable, potentially leading to an underestimation of the number of circulating CREC clusters and an incomplete understanding of their transmission dynamics. To gain a more comprehensive understanding of CREC epidemiology, routine and timely large-scale sequencing of CREC genomes across different regions is essential.

In conclusion, this study examines the worldwide distribution of CRECs and the global dissemination of commonly prevalent STs. We can effectively monitor and control infections by identifying infection reservoirs and analyzing genomic data to study the geographical transfer patterns and dynamics of CREC across countries, regions, and continents. This approach enables us to provide timely alerts to infection control decision-makers and support the implementation of effective strategies and policies to address CREC infections in various countries. Ultimately, this can help reduce the burden of resistance to antibiotics such as carbapenems.

## Methods

**Downloaded genomic sequences of CREC strains from the public database.** We obtained a dataset from the NCBI Pathogens database (https://www.ncbi.nlm.nih.gov/pathogens/) until March 1st, 2023. This dataset comprises a total of 11,051 CREC that possess carbapenem resistance genes (CRGs) such as "$bla_{KPC}$", "$bla_{NDM}$", "$bla_{OXA-48}$", "$bla_{OXA-232}$", "$bla_{OXA-181}$", "$bla_{IMP}$", or "$bla_{VIM}$". We assembled 2320 of these CRECs that did not have complete sequences. A total of 7918 isolates excluding strains lacking temporal and location information were included in the study (Supplementary Data 1). The species of the genomes were verified using SpeciesFinder v2.0[28]. All assembled genomic data were subjected to quality control measures, which involved filtering out data from collection sites and removing assemblies with genome sizes exceeding 6 Mbp or falling below 4 Mbp. The definition of carbapenem resistance consists of the presence of one of the carbapenemase-encoding genes mentioned above.

**Routine bioinformatic analysis.** The short-reads sequencing data from Illumina was processed using SPAdes v3.14.1 for assembly. Subsequently, genome annotations were conducted using Prokka 1.14.5 (https://github.com/tseemann/prokka). The allele profiles and sequence types (STs) of CREC strains were determined using MLST 2.23.0 (https://github.com/tseemann/mlst). The virulence genes and ARGs were detected using the VFDB and ABRicate databases (https://github.com/tseemann/abricate), respectively. The identification of virulence factors and ARGs was considered presence when their identity and coverage exceeded 90% and 70%, respectively. The assessment of phylogroups was performed using ClermonTyper, a tool widely used in the scientific community. This tool can distinguish seven phylogroups: A, B1, B2, C, D, E, and F (http://clermontyping.iame-research.center/)[43].

**Plasmid replicon identification and prediction of mobility typing.** The presence of plasmid replicons was determined using the PlasmidFinder software (v2.1.1) with default cutoffs (≥70% coverage and ≥90% identity)[44]. The potential plasmid-associated contigs of *E. coli* genomic sequences were investigated using the MOB-suite v3.0.3 database with the default settings (https://github.com/phac-nml/mob-suite). The MOB-cluster algorithm utilized single-link clustering and established a threshold of 0.05 Mash distance, equivalent to 95% of ANI[45]. Plasmid sequences in the CREC genome assemblies were identified using the MOB-recon tool, which utilized the MOB-suite v3.0.3 databases and default parameters[46,47]. The mobility of plasmids was predicted using the mob-typer tool in the MOB-suite software. Plasmids were classified into three categories: conjugative, mobilizable, and non-mobilizable. We designate conjugative and mobilizable plasmids as mobile plasmids, indicating plasmids with potential transferability potentials. By identifying resistance and virulence genes harbored on the mobile plasmid, those genes likely to be transferred with the plasmid can be predicted.

**Phylogenetic analysis.** An evolutionary tree analysis was performed on 17 major CREC STs that have been found to have worldwide transmission ranges across three to five continents. A comprehensive study was conducted on a total of 268 representative strains. This was achieved by randomly selecting one strain from each branch of each ST. The maximum-likelihood phylogenetic tree construction was performed using FastTree[48]. The trees were visualized using Figtree (tree.bio.ed.ac.uk/software/figtree) and annotated using iTOL v6[49]. The R package BactDating was utilized to perform Bayesian dating of the nodes of bacterial phylogenetic trees (https://github.com/xavierdidelot/BactDating). The fundamental methodology is based on the

findings of Didelot's study[50]. The fast trees and isolation dates of the strains were entered as input. Execute the command to assess the convergence of the Markov Chain Monte Carlo (MCMC) algorithm, specifically BactDating. Estimate the convergence using the standard parameters (https://xavierdidelot.github.io/BactDating/articles/yourData.html) provided on the reference webpage (Supplementary Note 1).

We utilized time-stamped phylogenetic trees and host ancestry reconstruction to analyze the temporal and locational data of CREC strains within a particular tree branch, referring to the methods of da Silva's study with some modifications[51]. We used time-based phylogenetic trees to investigate strains' sequence and spatial dynamics within a phylogenetic lineage. This allowed us to track strains' temporal and positional information on the same evolutionary branch. Additionally, we simulated the propagation trajectory of a CREC strain from its geographic origin to the neighboring area of a later-emerging CREC strain. The size of each arrow is proportionally adjusted based on the estimated number of transfers occurring between the countries. The dates provided represent the approximate initial transmission between each pair of countries. The time interval represents the expected spread duration between each pair of nations. This enabled us to differentiate between numerous routes of transmission. Intercontinental STs were classified as those with more than 10 CREC strains spreading across each continent. Similarly, countries that exhibited the presence of 5 or more strains along a specific transmission route were depicted on the map. In the case of countries displaying 1 to 4 disseminated strains, cross-continental transmission routes were deliberately included in the analysis, considering the number of strains and their timing. However, countries experiencing intra-state transmission that was either in close proximity or densely concentrated were not represented on the map. Key transmission hubs were designated as countries that have transmitted to five or more states and have transmitted more than ten CREC isolates (Supplementary Note 1).

**Statistics and reproducibility**. The data were statistically analyzed and visualized using GraphPad Prism 8.0 (GraphPad Prism Software, Inc.) and R 4.2.0 (Lucent Technologies, Jasmine Mountain, USA). The heat maps were presented using TBtools, as described in the study by Chen et al.[52]. The spatial distribution maps and propagation roadmaps of CRECs were generated using the "Pyecharts" package in Python 3. We created a geographical grid of coordinates representing potential origins of CRECs. Given the large number of propagation paths in the result, we introduced the main path data into the Python3 application Pyecharts. The map mainly shows the number of transmissible strains between countries greater than or equal to 5, while the number of transmissible strains between 1 and 4 is prioritized to show the routes of intercontinental transmission.

**Reporting summary**. Further information on research design is available in the Nature Portfolio Reporting Summary linked to this article.

## Data availability
The datasets supporting the conclusions of this article are available in the NCBI repository. A full list of accessions of data used in this study are provided in Supplementary Data 1. Supplementary Data 2–5 contain the source data for Fig. 1b–e; Supplementary Data 6–7 contain the source data for Fig. 2b, c; Supplementary Data 8 and 9 contain the source data for Figs. 3a and 4, respectively.

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

## Acknowledgements

This study was supported by the National Natural Science Foundation of China (Grant No. 32170141) and China Medical Board (no. 20-365).

## Author contributions

J.W.H. and Y.Z.Z. designed the study. J.W.H., C.L., M.L., Q.T.L. analyzed the data under the supervision of T.R., Y.F.C., P.H.N. and Y.Z.Z. JWH wrote the first draft of the manuscript, with contributions from Z.S., Y.N.Z., X.K.G. and Y.Z.Z. All authors read and approved the final manuscript.

## Competing interests

The authors declare no competing interests.
