## [Peer review file · Communications Biology]

Reviewers' comments:

Reviewer #1 (Remarks to the Author):

* Huang conducted an analysis of carbapenem-resistant Escherichia coli in public databases and erroneously labeled it as a global epidemiological study. This assertion is fundamentally incorrect. Without a systematic epidemiological design, it is inappropriate to categorize an analysis of a convenience dataset as an epidemiological study.

* This global dataset is undoubtedly subject to significant bias due to many of the different situations. Additionally, without proper strain selection, bioinformatics analysis could be influenced by clonal effects stemming from potential clonal expansion.

* Thirdly, I struggled to identify a clear definition of "spread" concerning these strains. Furthermore, the absence of real-world epidemiological connections renders the description of "national, international, and intercontinental spread" both meaningless and misleading.

* Furthermore, it is not possible to define "Intercontinental STs" or a "transmission route" solely based on the number of strains, given the potential for clonal expansion.

* Lastly, I found minimal evidence of conceptual advancements within this manuscript.

Reviewer #2 (Remarks to the Author):

Summary

There remains a gap in the understanding of the global transmission of CRE. The authors seek to explore this gap by characterizing the genetic diversity, antibiotic resistance and virulence profiles of 7918 CREC strains. They achieve this through simulating transmission dynamics of major sequence types across continents. The authors discovered major transmission hubs for three prevalent sequence types and modeled dates initial transmission events.

Major Comments

Although the study is extensive and comprehensive it needs more details on the methods- specifically the parameters used in the Bayesian modeling of transmission dynamics in order for it to be reproducible. One major concern- is the lack of discussion regarding the open genome of E. coli and the role of horizontal gene transfer may play in biases phylogenetic modeling. Details in the methods regarding the alignment used in phylogenetic modeling and how horizontal transmission was accounted for would benefit the reliability of the simulated transmission events. Otherwise- we may be looking at a transmission network rather than true vertical transmission and transmission events may be biased.

Minor Comments

As the authors also present results outside of CRE such as other classes of resistance and virulence, the introduction could also use more details on the significance of these features and why they are of importance.

Line by Line Reviews

Line 52- 1st should be "first"

Line 79- Clarify how you define mobile plasmid, either in the intro or in the text

Line 91- What do you mean by splice? Do you mean omit?

Line 94- Were these the short reads/unassembled or the assembled draft genomes?

Line 120- Clarify what you mean by comparing gene sequences with those found in mobile plasmids. Mobile plasmids is a broad term and does not always include resistance or virulence genes. Do you mean you used a reference?

Line 134- Reference is spelt wrong (should be de Silva)

Line 134- Specify the modifications. What parameters were used to generate these phylogenetic

trees?

Line 136- How did you define a strain? Perhaps clarify in methods

Line 138- How is a transmission route defined?

Line 156- Why did you focus on those with over 100 CREC strains (again how are you defining strains?)

Figure 1A- and 1C- Super cool figures!!

Line 219- Authors state an upward trend but this is not normalized to overall count which is also trending upwards.

Line 271- Transferable plasmid- needs to be defined either in intro or in this section.

Figure 2- How was "most prevalent" defined? By a certain count or percentage?

Line 293- "were to construct" should be "were used to construct."

Line 324- Again explain what is mobile plasmid vs conjugative vs transferable

Line 331- Spell out numbers when they start a sentence.

Line 336- Why is aminoglycosides capitalized but sulfonamide not?

Line 352- See above about spelling out numbers; start new paragraph because now discussing virulence.

Line 355- See above about spelling out numbers; also capitalize the virulence gene as your starting a new sentence.

Figure 6- might be more insightful to change x-axis of tree tighter so see more from recent transmission events.

Line 389- Need more info on the methods for the Bayesian modelling.

Reviewer #3 (Remarks to the Author):

I appreciate the authors' efforts to investigate the global dissemination patterns and dynamic trends of CREC isolates from 75 countries in this study. However, the abstract needs to be revised because it does not clearly state the goals of the study, the major conclusions, or how important they are for managing CREC infections. Also, I suggest the authors restructure the "discussion" to convey the information legibly and streamline the logic. Adding to it the following mechanical flaws were noticed that need to be addressed before publication in 'Nature Communications'.

- The study data are poorly interconnected to bring in the information on how the study outcome be extrapolated for prevention and control purposes. Eg. When authors stress on identification of transmission hubs, they may mention its significance like early detection of new outbreaks, and make targeted interventions like public awareness, sanitation, and vaccination as prevention and control measures.
- Unlike the authors' four-type classification, the classification of plasmid based on motility consists of three types; Conjugative, Mobilizable, and Unmobilizable. The authors may hence clarify whether the fourth type, conjugative and mobilizable is a term they have coined; if so on what basis?
- R package BactDating tracks the bacterial genealogy based on a dated phylogenetic tree. Since the authors have provided only the 7918 CREC isolates spanning from 2003 -2023 and have excluded WGS that had no information on the date of isolation, the authors may have to clarify how authoritative is the three transmission dates (1975, 1870, and 1931) that led of MDR ancestors of CRECs that carry plasmids harboring ESBL resistance genes.
- The coloring pattern of lineages enhances to tract of the transmission sources. To this line, the authors may please describe how the three vital host switches probably occurred far behind the dates 1975, 1870, and 1931. I suggest either a justification/reference or reconstruction of the phylogenetic tree in Fig.6C.
- The authors have highlighted the increasing dominance of NDM-5 and KPC-2. However, it lacks information on how the authors in the middle of NDM and KPC results bring in these variants within NDM and KPC.
- Line 97: E. coli genome size ranges from 4 to 6 Mb, where sizes above 5.5 are exclusively found in poultry isolates. Authors may explain why a cutoff of 4- 7 Mb was fixed to filter the genome data.
- Line 120: The Authors have identified the plasmid origin of virulence and antibiotic resistance,

however, the authors may explain how the plasmid origin of virulence and antibiotic resistance was identified while considering potential chromosomal insertions.

- Line 162: says "Strains with two or more CRGs are prevalent in India, Israel 163, and various African countries". as it appears that other continents, apart from South America, also have significant prevalence; Africa represents the most strains of this kind.
- Line 180-181: It says blaOXA48 is prevalent in humans, however, it does not correlate with Fig. 1E data where it is mentioned that NDM is exclusively high from human samples. Also, the authors may provide clarity on how the percentages represented in this Fig. 1E are calculated.
- In Lines 245-264, the authors have described the dynamics of CRGs in different countries. I suggest to interpret and highlight some commonalities in the CRG dynamics such like the increasing prevalence of NDM-5 over other CRGs.
- In lines 255 - 275, the authors convey the prevalence of ARGs and 11 other antibiotics. I would suggest the authors convey why this was done and how relevant it is to the dynamics or management of CREC.
- The authors have categorized the origin of the ARGs into chromosomes, and plasmids and have further mentioned the mobilizability of the plasmid. I insist the authors provide information of
- When the authors narrow down the investigation to specific STs, like ST167, ST131, and ST140, they may stress its importance and provide appropriate justification. Or, I feel these must be conveyed legibly. The same applies to the study of the prevalence and dynamics of different virulence genes.
- In total the authors may summarize the distribution of the significantly prevalent ST against its Phylogroups, and then about its country/continent-wise prevalence followed by its transmission risk and transmission hubs. Such a graphical summary would convey the whole effort of this work that aims to give a global picture of the prevalence, dynamics, and transmission potency of CRECs across the Globe.

Response to Reviewer #1:

Q1: Huang conducted an analysis of carbapenem-resistant *Escherichia coli* in public databases and erroneously labeled it as a global epidemiological study. This assertion is fundamentally incorrect. Without a systematic epidemiological design, it is inappropriate to categorize an analysis of a convenience dataset as an epidemiological study.

R: Thank you so much for your comments. We agree with your comments. In general, epidemiologic studies require rigorous and standardized experimental designs. We consider changing the title to: “Genomic Investigation of Global Carbapenem-Resistant *Escherichia coli* Reveals Diverse Spatiotemporal Epidemiological Characteristics.”

The advancements in genomics and sequencing technology have led to the gradual integration of NGS technologies into epidemiological research and efforts for prevention and control. These technologies and the vast amount of data collected have significantly reduced the need for extensive field investigations previously necessary in traditional epidemiological studies. The utilization of diverse databases has significantly increased the availability of clinical and laboratory research resources. Epidemiological studies on COVID-19 have increasingly utilized publicly available sequencing data for investigating virus transmission, evolution, and related aspects. This reliance stems from disease and data management variations across national regions ^{1, 2, 3}. Several genomic epidemiology studies have been conducted on global Carbapenemase-Producing *E. coli*, including smaller-scale studies by Peirano ⁴ and Boutzoukas ⁵.

Although there may be bias in our data inclusion and selection, our research is valuable for quickly identifying new outbreaks and implementing effective interventions such as public education efforts, improved hygiene practices, and vaccination strategies. Our findings will enhance the understanding of the response to various carbapenem-resistance genes in *E. coli* strains when exposed to different combinations of antibiotics. Our findings contribute to future experimental studies on the spread pattern of *E. coli* ⁶.

Q2: This global dataset is undoubtedly subject to significant bias due to many of the different situations. Additionally, without proper strain selection, bioinformatics analysis could be influenced by clonal effects stemming from potential clonal expansion.

R: Thank you for your comment. Indeed, excluding clonally expanded strains from public sequence databases is a challenging task. To mitigate the impact of these strains, we employed several strategies. Firstly, we constructed a phylogenetic tree using Snippy, excluding potential clonal expansions as suggested by previous studies ^{7, 8}. Additionally, we restricted the number of strains considered during the analysis of geographical spreading.

Q3: Thirdly, I struggled to identify a clear definition of "spread" concerning these strains. Furthermore, the absence of real-world epidemiological connections renders the description of "national, international, and intercontinental spread" both meaningless and misleading.

R: Yes. The genetic distances between the various strains were obtained directly from the extensive collection of sequences in the public database. This study suggests that the presence of identical ST resistant strains in locations that are geographically distant but close in time may be attributed to geographical transfer. Wu ⁹ classified it as "Temporal and spatial transmission." The R package BactDating ¹⁰ was employed for inferring bacterial phylogenetic trees through Bayesian inference. The package can be found at <https://github.com/xavierdidelot/BactDating>. The methodology is based on a paper by Didelot ¹¹. We constructed time trees of evolution to track strains' temporal and positional information on the same branch. This allowed us to model the spread pathway, where a strain that appeared early spreads to a strain that appeared later in a different geographic location. This approach has also been utilized to demonstrate geographical transfers in other studies ^{11, 12}.

Q4: Furthermore, it is not possible to define "Intercontinental STs" or a

"transmission route" solely based on the number of strains, given the potential for clonal expansion.

R: It is important to acknowledge the objective existence of clonal expansion. In traditional epidemiologic investigations, clonal expansion cannot be entirely ruled out, even when each patient corresponds to a single strain.

In the revised manuscript, we clarified that the SNP locus method was employed to mitigate the impact of clonal expansion ^{7, 8}. While the number of strains does not directly determine transmission, limiting the number of strains can effectively mitigate the impact of certain clonal expansion strains.

Q5: Lastly, I found minimal evidence of conceptual advancements within this manuscript.

R: Thank you for your rigorous review. We apologize for the overall dissatisfaction you have expressed regarding our work. The motivation for this study stems from the limitations of previous small-scale studies in examining the spread of strains and resistance genes across larger geographic areas and longer periods. These limitations include sample size and the spatial span of sample time, which are challenging to address using conventional research methods ¹³. Hence, by examining a greater number and range of strains and studying the dissemination of strains and resistance genes, this research is expected to enhance traditional epidemiological studies significantly. Furthermore, as more strains are sequenced in the future, this work will continue to be refined and improved.

Reference

1. Xu B, Kraemer MUG. Open access epidemiological data from the COVID-19 outbreak. *Lancet Infect Dis* **20**, 534 (2020).
2. Lemieux JE, *et al.* Phylogenetic analysis of SARS-CoV-2 in Boston highlights the impact of superspreading events. *Science* **371**, (2021).

3. Brown CM, *et al.* Outbreak of SARS-CoV-2 Infections, Including COVID-19 Vaccine Breakthrough Infections, Associated with Large Public Gatherings - Barnstable County, Massachusetts, July 2021. *MMWR Morb Mortal Wkly Rep* **70**, 1059-1062 (2021).
4. Peirano G, *et al.* Genomic Epidemiology of Global Carbapenemase-Producing *Escherichia coli*, 2015-2017. *Emerg Infect Dis* **28**, 924-931 (2022).
5. Boutzoukas AE, *et al.* International epidemiology of carbapenemase-producing *Escherichia coli*. *Clin Infect Dis*, (2023).
6. Senghore M, *et al.* Inferring bacterial transmission dynamics using deep sequencing genomic surveillance data. *Nat Commun* **14**, 6397 (2023).
7. Yang C, *et al.* Outbreak dynamics of foodborne pathogen *Vibrio parahaemolyticus* over a seventeen year period implies hidden reservoirs. *Nature Microbiology* **7**, 1221-1229 (2022).
8. Stimson J, Gardy J, Mathema B, Crudu V, Cohen T, Colijn C. Beyond the SNP Threshold: Identifying Outbreak Clusters Using Inferred Transmissions. *Molecular Biology and Evolution* **36**, 587-603 (2019).
9. Wu Y, *et al.* Global Phylogeography and Genomic Epidemiology of Carbapenem-Resistant blaOXA-232-Carrying *Klebsiella pneumoniae* Sequence Type 15 Lineage. *Emerging Infectious Diseases* **29**, (2023).
10. Nascimento FF, Reis MD, Yang Z. A biologist's guide to Bayesian phylogenetic analysis. *Nat Ecol Evol* **1**, 1446-1454 (2017).
11. Didelot X, Croucher NJ, Bentley SD, Harris SR, Wilson DJ. Bayesian inference of ancestral dates on bacterial phylogenetic trees. *Nucleic Acids Res* **46**, e134 (2018).

12. da Silva KE, *et al.* The international and intercontinental spread and expansion of antimicrobial-resistant *Salmonella* Typhi: a genomic epidemiology study. *Lancet Microbe* **3**, e567-e577 (2022).

13. Effelsberg N, *et al.* Global Epidemiology and Evolutionary History of *Staphylococcus aureus* ST45. *J Clin Microbiol* **59**, (2020).

Response to Reviewer #2:

Summary

There remains a gap in the understanding of the global transmission of CRE. The authors seek to explore this gap by characterizing the genetic diversity, antibiotic resistance and virulence profiles of 7918 CREC strains. They achieve this through simulating transmission dynamics of major sequence types across continents. The authors discovered major transmission hubs for three prevalent sequence types and modeled dates initial transmission events.

Major Comments

Q1: Although the study is extensive and comprehensive it needs more details on the methods- specifically the parameters used in the Bayesian modeling of transmission dynamics in order for it to be reproducible. One major concern- is the lack of discussion regarding the open genome of E. coli and the role of horizontal gene transfer may play in biases phylogenetic modeling. Details in the methods regarding the alignment used in phylogenetic modeling and how horizontal transmission was accounted for would benefit the reliability of the simulated transmission events. Otherwise- we may be looking at a transmission network rather than true vertical transmission and transmission events may be biased.

R: Thank you for your critical review. We agree with your comments and have made revisions to the manuscript, incorporating parametric details and a discussion of bias. The BactDating R package was used for Bayesian dating of bacterial phylogenetic tree nodes (source: <https://github.com/xavierdidelot/BactDating>). The fundamental methodology is based on the work of Didelot ¹. The fasttrees and isolation dates of the strains were entered as input. Execute the command to assess the convergence of the Markov Chain Monte Carlo (MCMC) algorithm employed in BactDating. Estimate the convergence using the standard parameters on the reference webpage (<https://xavierdidelot.github.io/BactDating/articles/yourData.html>).

Our study was conducted to address the challenges encountered in larger studies of widespread transmission of bacteria and resistance genes, which were identified in previous smaller studies. The main challenge involved the collection of strains from diverse sources. Therefore, we investigated the feasibility of utilizing data from existing public databases, such as NCBI, to overcome this difficulty due to this dataset's substantial and expanding nature.

Nevertheless, it is evident from your comment that we have not adequately addressed the issue of "bias" in our research. Our models cannot accurately establish whether the strains and gene transmissions are interregional or vertical clonal expansions. The *E. coli* sequences in our database primarily originate from epidemiological studies conducted in various geographical regions, and case reports documenting specific infections. Considering the shared objective of both studies to minimize clonal expansion of strains, the resulting data bias from such amplification is relatively minimal. We restricted the number of strains analyzed and the examination of gene transfer across distant continents. This measure could reduce clonal amplification and enhance the study's credibility.

The specific commands and parameters are listed below:

*First input the phylogenetic tree, which can be loaded from a Newick file using the commands from the ape package: `t=read.tree('filename.nwk')`. Then calculate `sum(t$edge.length)` to determine if the branch length measurement is correct. Below 1 it needs to be adjusted using the command `t$edge.length=t$edge.length*L`, where *L* is the number of sites which were used to build this tree. Because this should be the total number of substitutions throughout the tree, so if we get a value below one or even only a bit above one, our branch lengths are probably not in the correct unit. The second required input the dates at which the isolates were sampled. We used the `decimal_date` function of the `lubridate` package to convert the other date formats to decimal years. All strain names entered `names(d)` correspond to the isolation dates entered previously. Then run `BactDating` using the main command:
Find an optimal root using: `rooted=initRoot(t,d)`*

Resolving Multiple Groupings: `rooted=multi2di(rooted)`

Function Reference URL: <https://rdr.io/cran/ape/man/multi2di.html>

Evaluating the Strength of the Time Signal Using Root-to-Tip Linear Regression: `r=roottotip(rooted,d)`

Prediction with BactDating's MCMC algorithm: `res=bactdate(rooted,d)`

Check that the traces of the parameter look stable: `plot(res,'treeCI')`

After constructing the time trees, we tracked the time and location information of CRECs on the same branch. Then we modeled the path of diffusion from the geographic locations of early-emerging strains to the geographic locations of late-emerging strains².

We customized the algorithm for obtaining propagation paths from evolutionary trees: mapping functions:

$f(\text{treenodes}) \rightarrow \text{collection of (source country, target country, number, date range)}$

input:

BactDating phylogeny tree

output:

a collection of propagation path with source country, target country, number and date range (in years).

Algorithm:

Identify the similar genes on the same branch (if only one, search the parent branches), figure out the meta path of the gene with the earliest date as source country and the one with the latest date as the target country. Then merge the meta paths with the same source and target into one propagation path. The number of the propagation path is the summary of these meta paths. The date range of the propagation path is the from the earliest year of the meta path to the latest year of the meta path.

Notice: the meta paths with the same countries but the source and the target are reverse are considered as different propagation paths. In the graph they are displayed as two curves in the opposite direction.

We created a geographical grid of coordinates representing potential origins of CRECs.

Given the large number of propagation paths in the result, we introduced the main path data into the Python3 application Pyecharts. The map mainly shows the number of transmissible strains between countries greater than or equal to 5, while the number of transmissible strains between 1 and 4 is prioritized to show the routes of intercontinental transmission.

Minor Comments

Q2: As the authors also present results outside of CRE such as other classes of resistance and virulence, the introduction could also use more details on the significance of these features and why they are of importance.

R: Thank you for your comments, the manuscript has been revised accordingly. Certain gram-negative bacilli, such as *E. coli*, exhibit high levels of drug resistance and possess virulence-associated genes. The presence of multidrug-resistant bacteria indicates that the development of various antibiotic resistance genes, such as carbapenem resistance, in bacteria is not necessarily unrelated but rather interconnected through a series of associations. Hence, it is important to monitor the resistance capacity of major antibiotic types to understand the distribution pattern of carbapenem-resistant *E. coli*, particularly the presence of multidrug-resistant genes.

The virulence profiles of CREC can exhibit variability, with certain strains harboring factors that augment their capacity to induce severe infections. Knowledge of these virulence factors is crucial for evaluating risk and making informed clinical decisions. The introduction provides a more detailed description of the significance and importance of these additional classes of resistance and virulence traits in the third paragraph of the introduction.

Line by Line Reviews

Q3: Line 52- 1st should be “first”

R: Thank you for your correction. This word has been corrected.

Text modified: Line 54.

Q4: Line 79- Clarify how you define mobile plasmid, either in the intro or in the text

R: We have added definitions of these concepts to the text and methods.

Conjugative plasmids contain transfer genes, also called *tra* genes, that encode the machinery required for the transfer process. The presence of these genes in the plasmid facilitates interbacterial transfer, leading to the dissemination of diverse traits, including antibiotic resistance, virulence factors, and metabolic capabilities³. Mobilizable plasmids do not possess the full complement of genes necessary for autonomous intercellular transfer. However, they can depend on other conjugative components, such as conjugative plasmids or integrating conjugative elements (ICEs), to aid in their transfer. This study designates both conjugative and mobilizable plasmids as mobile plasmids, indicating their inherent ability to transfer.

Text modified: Line 86~97.

Q5: Line 91- What do you mean by splice? Do you mean omit?

R: No, it should be the word “assemble” here. That means, we assembled the 2320 CRECs that don't have complete sequences.

Text modified: Line 117.

Q6: Line 94- Were these the short reads/unassembled or the assembled draft genomes?

R: They were the draft genomes assembled by ourselves.

Q7: Line 120- Clarify what you mean by comparing gene sequences with those found in mobile plasmids. Mobile plasmids is a broad term and does not always include resistance or virulence genes. Do you mean you used a reference?

R: Sorry, that sentence in the article didn't express clearly. We have added definitions of “mobile plasmids” in Introduction.

We employed the mob-typer module within the MOB-suite software to classify

plasmids into three categories based on their mobility: conjugative, mobilizable, and non-mobilizable. Conjugative and mobilizable plasmids are collectively called mobile plasmids due to their ability to be transferred between cells. Identifying resistance and virulence genes on mobile plasmids allows predicting genes likely to be transferred along with the plasmid. If the current definition is deemed inappropriate, we can substitute "mobile plasmid" with "conjugative and mobilizable plasmids" throughout the paper. The VFDB and ABRicate databases were utilized to identify virulence genes and antibiotic resistance genes (ARGs) in the sample. The ABRicate database can be accessed at <https://github.com/tseemann/abricate>.

Mobile plasmid-carried genes exhibit greater transferability compared to chromosome-carried genes. Consequently, resistance genes located on plasmids have varying capacities for horizontal transfer, leading to different risks of resistance transmission⁴. Our study identified the resistant and virulence gene profiles on various mobile plasmids of CREC. This finding is significant for monitoring and managing the transmission risk.

Text modified: Line 86~97.

Q8: Line 134- Reference is spelt wrong (should be de Silva)

Line 134- Specify the modifications. What parameters were used to generate these phylodynamic trees?

R: Thanks for the advice, we have rechecked that reference, it should be “da Silva”).

For the phylogenetic tree, we refer to da Silva's article².

Text modified: Line 168.

The parameters we used was showed in the response to Q1.

Q9: Line 136- How did you define a strain? Perhaps clarify in methods

R: Strains, as defined here, are independent strains that are separate and do not have a direct clonal amplification relationship with each other. We refer to Stimson's and Yang's articles^{5,6} to define a "strain" by limiting SNPs.

Q10: Line 138- How is a transmission route defined?

R: The prediction derived from the phylogenetic tree is commonly referred to as a "geographical spread." We consulted da Silva's publication "The International and Intercontinental Spread and Expansion of Antimicrobial-resistant Salmonella Typhi: A Genomic Epidemiology Study" ². Our study defines "a transmission route" as a unidirectional path through which transmission occurs between two distinct locations. The size of each arrow is proportional to the estimated volume of transmissions between the countries. The dates provided represent the estimated initial spread between each pair of countries. The time interval represents the expected duration of the spread between each pair of nations. Transmission refers to the geographic spread.

Q11: Line 156- Why did you focus on those with over 100 CREC strains (again how are you defining strains?)

R: The extensive diversity of ST types makes it challenging to fully understand the distribution and transmission of these types across various geographic regions. Strains exceeding 100 in number can be classified as major sequence types (STs), which possess greater representativeness. Studying the geographical transmissions of these major STs is of greater significance.

Q12: Figure 1A- and 1C- Super cool figures!!

R: Thank you very much!

Q13: Line 219- Authors state an upward trend but this is not normalized to overall count which is also trending upwards.

R: In Figure 2B, we have showed the percentages of various ST types, ST167, ST361, ST219, and ST648 which are ST types that are trending upward from 2013-2023, which can reflect the increase in prevalence.

Q14: Line 271- Transferable plasmid- needs to be defined either in intro or in this section.

R: Here “Transferable plasmid” means the same as “mobile plasmid” in Figure 5. We have defined “mobile plasmid” in updated Introduction. Both conjugative and mobilizable plasmids are designated as mobile plasmids in this study, signifying plasmids possessing an inherent propensity to transfer.

Text modified: Line 86~97.

Q15: Figure 2- How was “most prevalent” defined? By a certain count or percentage?

R: In Figure 2, the total number of bacteria we studied is consistent and therefore can be interpreted as having the highest percentage.

Q16: Line 293- “were to construct” should be “were used to construct.”

R: Thank you very much for your suggestion, it has been corrected.

Text modified: Line 333.

Q17: Line 324- Again explain what is mobile plasmid vs conjugative vs transferable

R: The tool MOB-recon and the clustered plasmid reference databases of MOB-suite⁷ were utilized to reconstruct individual plasmid sequences from draft genome assemblies. The chromosomal and plasmid sequences of 7918 CRECs were isolated using mob-suite. Virulence and resistance genes were identified in distinct chromosomal and plasmid sequences (Table S1, S2). The mob-typer was employed for the prediction of plasmid transposability. Additionally, the positions of predicted mobilizable and conjugative plasmid initiation were compared with the positions of virulence and resistance sequence genes. The ability of CREC plasmids to disseminate virulence and resistance genes can be predicted using the findings from Table S3 and S4. Conjugative plasmids contain transfer genes, also known as tra genes, that encode the machinery required for the transfer process. The presence of these genes in the plasmid facilitates inter-bacterial transfer, thereby facilitating the dissemination of diverse traits, including antibiotic resistance, virulence factors, and metabolic

capabilities³.

Conjugative plasmids facilitate the transmission of antibiotic resistance genes, enabling their swift dissemination within bacterial communities. The emergence and spread of multidrug-resistant bacteria pose a substantial obstacle to the efficacy of antibiotics in infection treatment⁸.

A mobilizable plasmid is a genetic element present in bacteria that lacks the necessary genes for self-transfer between cells. Mobilizable plasmids require the presence of other conjugative elements, such as conjugative plasmids or integrating conjugative elements (ICEs), to aid in their transfer. In this study, both conjugative and mobilizable plasmids are classified as mobile plasmids due to their inherent ability to transfer.

Q18: Line 331- Spell out numbers when they start a sentence.

R: Thank you for the suggestion. We updated it.

Text modified: Line 392, 395.

Q19: Line 336- Why is aminoglycosides capitalized but sulfonamide not?

R: Thank you for your comment, we corrected it.

Text modified: Line 366.

Q20: Line 352- See above about spelling out numbers; start new paragraph because now discussing virulence.

R: Thank you for the comment, we changed “66.79%” to “about sixty-seven percent”.

Text modified: Line 392.

Q21: Line 355- See above about spelling out numbers; also capitalize the virulence gene as your starting a new sentence.

R: Thanks for the correction, we have changed the letters and numbers at the beginning of the sentence.

Text modified: Line 395, 396.

Q22: Figure 6- might be more insightful to change x-axis of tree tighter so see more from recent transmission events.

R: Yes, we updated Fig 6.

Q23: Line 389- Need more info on the methods for the Bayesian modelling.

R: We utilized the BactDating method described in the article "Bayesian inference of ancestral dates on bacterial phylogenetic trees" by Didelot ¹. This approach utilizes Bayesian estimation to determine the molecular clock and coalescent rates. The specific parameters, modeling, and algorithmic information have been addressed in Q1.

Reference

1. Didelot X, Croucher NJ, Bentley SD, Harris SR, Wilson DJ. Bayesian inference of ancestral dates on bacterial phylogenetic trees. *Nucleic Acids Res* **46**, e134 (2018).
2. da Silva KE, *et al.* The international and intercontinental spread and expansion of antimicrobial-resistant Salmonella Typhi: a genomic epidemiology study. *Lancet Microbe* **3**, e567-e577 (2022).
3. Dimitriu T, Matthews AC, Buckling A. Increased copy number couples the evolution of plasmid horizontal transmission and plasmid-encoded antibiotic resistance. *Proc Natl Acad Sci U S A* **118**, (2021).
4. Smillie C, Garcillán-Barcia MP, Francia MV, Rocha EP, de la Cruz F. Mobility of plasmids. *Microbiol Mol Biol Rev* **74**, 434-452 (2010).
5. Yang C, *et al.* Outbreak dynamics of foodborne pathogen *Vibrio parahaemolyticus* over a seventeen year period implies hidden reservoirs. *Nature Microbiology* **7**, 1221-1229 (2022).

6. Stimson J, Gardy J, Mathema B, Crudu V, Cohen T, Colijn C. Beyond the SNP Threshold: Identifying Outbreak Clusters Using Inferred Transmissions. *Molecular Biology and Evolution* **36**, 587-603 (2019).

7. Robertson J, Nash JHE. MOB-suite: software tools for clustering, reconstruction and typing of plasmids from draft assemblies. *Microb Genom* **4**, (2018).

8. Sher AA, *et al.* Conjugative RP4 Plasmid-Mediated Transfer of Antibiotic Resistance Genes to Commensal and Multidrug-Resistant Enteric Bacteria In Vitro. *Microorganisms* **11**, (2023).

Response to Reviewer #3:

Q1: I appreciate the authors' efforts to investigate the global dissemination patterns and dynamic trends of CREC isolates from 75 countries in this study. However, the abstract needs to be revised because it does not clearly state the goals of the study, the major conclusions, or how important they are for managing CREC infections. Also, I suggest the authors restructure the "discussion" to convey the information legibly and streamline the logic. Adding to it the following mechanical flaws were noticed that need to be addressed before publication in 'Nature Communications'.

R: We apologize for any shortcomings in the clarity of our study's Abstract and Discussion sections, which may have hindered the communication of the study's aims, findings, and importance. Based on your comments, we have revised the abstract and discussion section.

Our study aimed to examine the distribution and evolutionary traits of important resistance genes in *E. coli*. We employed extensive genomic data on carbapenem-resistant bacteria to accomplish this goal. Furthermore, we investigated the worldwide distribution of the bacteria and the resistance genes they are associated with. Our study reveals intercontinental transmission of major CREC strains, with certain locations playing a crucial role in facilitating this dissemination. This discovery has significant implications for the efficient and targeted management of infection and transmission of CREC and other highly resistant bacteria.

Q2: The study data are poorly interconnected to bring in the information on how the study outcome be extrapolated for prevention and control purposes. Eg. When authors stress on identification of transmission hubs, they may mention its significance like early detection of new outbreaks, and make targeted interventions like public awareness, sanitation, and vaccination as prevention and control measures.

R: Thank you very much for your comments. Your comment has been included in the revised draft. Distinct strategies for prevention and treatment are required for various

strains of CREC and multidrug-resistant *E. coli*. Efficient and timely interventions are necessary to control the spread of drug-resistant bacteria and genes, as indicated by the identified spread centers in this study.

Text modified: Line 609~610, 619~621.

Q3: Unlike the authors' four-type classification, the classification of plasmid based on motility consists of three types; Conjugative, Mobilizable, and Unmobilizable. The authors may hence clarify whether the fourth type, conjugative and mobilizable is a term they have coined; if so on what basis?

R: The "mobile plasmid" does not belong to the fourth category. We employed the mobtyper tool within the MOB-suite software to classify plasmids into three categories based on their mobility: conjugative, mobilizable, and non-mobilizable. Conjugative and mobilizable plasmids are commonly referred to as "mobile plasmids" due to their ability to be transferred between cells. If the current definition is deemed unsuitable, the term "mobile plasmid" can be replaced with "conjugative and mobilizable plasmids" throughout the paper. The identification of resistance and virulence genes present on mobile plasmids allows for the prediction of genes that are likely to be transferred along with the plasmid.

Q4: R package BactDating tracks the bacterial genealogy based on a dated phylogenetic tree. Since the authors have provided only the 7918 CREC isolates spanning from 2003 -2023 and have excluded WGS that had no information on the date of isolation, the authors may have to clarify how authoritative is the three transmission dates (1975, 1870, and 1931) that led of MDR ancestors of CRECs that carry plasmids harboring ESBL resistance genes.

R: Yes, we employed the method of BactDating to estimate branch points and make temporal predictions within the evolutionary tree. The prediction relies on the genetic distance between strains and the specific time of strain collection. This study only included strains that had a specified collection time. Strains lacking precise collection timestamps have been omitted. The limited collection time of the sequenced

carbapenem-resistant *E. coli* results in predictions that extend significantly beyond the software entry time. The absence of data on strains with precise sampling dates from earlier periods has a negative impact on the accuracy of predicted time points. Thus, the software provides a variety of confidence intervals. To improve the precision of time nodes, it is necessary to incorporate additional strains and genomic data ¹.

Q5: The coloring pattern of lineages enhances to tract of the transmission sources. To this line, the authors may please describe how the three vital host switches probably occurred far behind the dates 1975, 1870, and 1931. I suggest either a justification/reference or reconstruction of the phylogenetic tree in Fig.6C.

R: BactDating is a recently developed predictive methodology employed in several published articles ². The methodology was also utilized in a prior study ³. The study predicts the timing of evolutionary branching nodes over a range of years. As depicted in Figure 6C, the predicted occurrence times are approximately 1975 (1941-1998), 1931 (1838-1992), and 1870 (1775-1939). The availability of previously sequenced strains can enhance BactDating results, even in cases where the collection times of the strains are not accurately recorded.

Yebra provides an additional reference for consideration. ⁴. BactDating was specifically utilized in the context of *S. aureus*, was employed to forecast the timing of epidemiological occurrences.

Fig. 2. Contemporary bovine *S. aureus* originated from human-to-bovine host switches that occurred in the last 2,500 y. Time-scaled tree of the global, bovine-enriched *S. aureus* dataset ($n = 3,915$) with host ancestral reconstruction. The maximum-likelihood tree was generated with IQ-Tree2, dated using BactDating, and host ancestral states were inferred using SIMMAP. Branches are colored according to the host yielding the highest probability. Main CCs are collapsed in triangles and colored according to their reconstructed original host. Nodes supported by 100% bootstrap are indicated by an asterisk. Key nodes are annotated with their predicted most recent common ancestors (in years before present). Arrows point to the main evolutionary jumps into bovine population.

Fig BactDating used in Yebra's study

Q6: The authors have highlighted the increasing dominance of NDM-5 and KPC-2. However, it lacks information on how the authors in the middle of NDM and KPC results bring in these variants within NDM and KPC.

R: Both NDM and KPC are significant antibiotic-resistance gene complexes, and their various types play a crucial role in epidemiologic studies and the prevention and control of resistance. The relevant information has been included in the revised manuscript.

Text modified: Line 217~218.

Q7: Line 97: *E. coli* genome size ranges from 4 to 6 Mb, where sizes above 5.5 are exclusively found in poultry isolates. Authors may explain why a cutoff of 4- 7 Mb was fixed to filter the genome data.

R: When the assembled genome size of *E. coli* is less than 4 Mb, the results indicate that low-quality sequencing data and much more genomic fragments during the sequencing procedure may be lost and result in an incomplete representation of the genome. A genome size of more than 6 Mb suggests potential sequencing

contamination.

Text modified: Line 122.

Q8: Line 120: The Authors have identified the plasmid origin of virulence and antibiotic resistance, however, the authors may explain how the plasmid origin of virulence and antibiotic resistance was identified while considering potential chromosomal insertions.

R: The plasmid and chromosome segments of the genome were separated using Mob-recon. Resistance genes were screened and the plasmid type was predicted separately. Strains carrying resistance genes on conjugative and mobilizable plasmids were considered to have a higher capacity for resistance transfer and transmission. If there are no identifiable upstream or downstream sequences of a horizontally transferred gene in the chromosome, it can be concluded that the gene did not originate from a chromosomal insertion ^{5,6}.

Q9: Line 162: says "Strains with two or more CRGs are prevalent in India, Israel, and various African countries". as it appears that other continents, apart from South America, also have significant prevalence; Africa represents the most strains of this kind.

R: Yes, Epidemiological surveys have shown that phylogroup B1 strains are prevalent in Africa and multiple European countries, contributing significantly to CRECs ⁷.

Q10: Line 180-181: It says blaOXA48 is prevalent in humans, however, it does not correlate with Fig. 1E data where it is mentioned that NDM is exclusively high from human samples. Also, the authors may provide clarity on how the percentages represented in this Fig. 1E are calculated.

Q10: In Figure 1B, we conducted a separate analysis to determine the percentage of sample sources in various CREC resistance genes. Additionally, a comparison was made between the rates of identical genes across various sample sources. This result has multiple possible interpretations. The study found that *bla*_{NDM} has the highest

prevalence of resistance genes in CREC originating from human sources. Additionally, the percentage of the resistance gene *bla*_{OXA-48} in human origins is more significant than its prevalence in environmental and animal sources. Given the considerable variation in the absolute number of strains across different sources, examining the relative proportion of identical resistance genes among various sample sources is intriguing.

Q11: In Lines 245-264, the authors have described the dynamics of CRGs in different countries. I suggest to interpret and highlight some commonalities in the CRG dynamics such like the increasing prevalence of NDM-5 over other CRGs.

R: Thank you for your advice. The increasing prevalence of NDM-5 has been a prominent characteristic of CREC resistance genes for more than ten years. This is clarified in our updated manuscript.

Text modified: Line 298~300.

Q12: In lines 255 - 275, the authors convey the prevalence of ARGs and 11 other antibiotics. I would suggest the authors convey why this was done and how relevant it is to the dynamics or management of CREC.

R: Thank you for your comments. Certain gram-negative bacilli, such as *E. coli*, exhibit high levels of drug resistance and possess virulence-associated genes. Hence, investigating resistance gene levels and transfer is important in preventing and controlling drug-resistant bacteria. The presence of multidrug-resistant bacteria indicates that the development of various antibiotic resistance genes, such as carbapenem resistance, in bacteria is not necessarily unrelated but interconnected in a cascade of associations. Hence, it is important to monitor the resistance capabilities of major antibiotics to understand the distribution pattern of carbapenem-resistant *E. coli*, particularly the presence of multidrug-resistant genes. The Introduction has been revised based on your comments and suggestions.

Text modified: Line 59~66.

Q13: The authors have categorized the origin of the ARGs into chromosomes, and

plasmids and have further mentioned the mobilizability of the plasmid. I insist the authors provide information of

R: We utilized the mob-typer module within the MOB-suite software to predict plasmid mobility. The classification of conjugative, mobilizable, and non-mobilizable elements can be determined using the mob-typer method, as described by Matlock and Williams^{8, 9}. Castañeda-Barba found that plasmid-borne resistance genes exhibit greater dissemination potential than chromosomal resistance genes¹⁰. The conjugative plasmid can self-transfer and exhibit the highest gene transfer capacity. The mobilizable plasmid contains the recognition site of the conjugative plasmid, resulting in a greater gene transfer ability than the non-mobilizable plasmid, which lacks this site. The analysis in mob-suite indicates that the presence of both conjugative and mobilizable plasmids in the same bacteria leads to a stronger gene transmission.

Q14: When the authors narrow down the investigation to specific STs, like ST167, ST131, and ST140, they may stress its importance and provide appropriate justification. Or, I feel these must be conveyed legibly. The same applies to the study of the prevalence and dynamics of different virulence genes.

R: We have chosen to focus on ST167, ST131, and ST140 in our paper due to their high prevalence among CRECs. Additionally, previous studies have shown that these strains have the ability to carry and transmit virulence or resistance genes, making them clinically significant threats^{11, 12, 13}.

Text modified: Line 332.

Q15: In total the authors may summarize the distribution of the significantly prevalent ST against its Phylogroups, and then about its country/continent-wise prevalence followed by its transmission risk and transmission hubs. Such a graphical summary would convey the whole effort of this work that aims to give a global picture of the prevalence, dynamics, and transmission potency of CRECs across the Globe.

R: We added a graphical summary for this study.

Phylogroups and spatiotemporal distribution of prevalent STs of CREC

National and intercontinental prevalence of CREC carbapenem resistance genes

Risk of transmission of CREC conjugative and mobilizable plasmids carrying resistance genes

Geographic transfer patterns of CREC and transmission hubs

Reference

1. Feng Y, *et al.* Key evolutionary events in the emergence of a globally disseminated, carbapenem resistant clone in the Escherichia coli ST410 lineage. *Communications Biology* 2, 322 (2019).

2. Guo Y, *et al.* Clinical and Microbiological Characteristics of *Mycobacterium kansasii* Pulmonary Infections in China. *Microbiol Spectr* **10**, e0147521 (2022).
3. Huang J, *et al.* Genetic diversity, antibiotic resistance, and virulence characteristics of *Staphylococcus aureus* from raw milk over 10 years in Shanghai. *International Journal of Food Microbiology* **401**, (2023).
4. Yebra G, *et al.* Multiclonal human origin and global expansion of an endemic bacterial pathogen of livestock. *Proc Natl Acad Sci U S A* **119**, e2211217119 (2022).
5. Arnold BJ, Huang IT, Hanage WP. Horizontal gene transfer and adaptive evolution in bacteria. *Nature Reviews Microbiology* **20**, 206-218 (2022).
6. Oliveira PH, Touchon M, Cury J, Rocha EPC. The chromosomal organization of horizontal gene transfer in bacteria. *Nature Communications* **8**, 841 (2017).
7. Stoppe N, *et al.* Worldwide Phylogenetic Group Patterns of *Escherichia coli* from Commensal Human and Wastewater Treatment Plant Isolates. *Frontiers in Microbiology* **8**, (2017).
8. Williams DJ, *et al.* The genus *Serratia* revisited by genomics. *Nature Communications* **13**, 5195 (2022).
9. Matlock W, *et al.* Genomic network analysis of environmental and livestock F-type plasmid populations. *The ISME Journal* **15**, 2322-2335 (2021).
10. Castañeda-Barba S, Top EM, Stalder T. Plasmids, a molecular cornerstone of antimicrobial resistance in the One Health era. *Nature Reviews Microbiology*, (2023).
11. Chen Q, *et al.* Characterization of blaNDM-5-and blaCTX-M-199-Producing ST167

Escherichia coli Isolated from Shared Bikes. *Antibiotics* **11**, 1030 (2022).

12. Morales Barroso I, López-Cerero L, Navarro MD, Gutiérrez-Gutiérrez B, Pascual A, Rodríguez-Baño J. Intestinal colonization due to Escherichia coli ST131: risk factors and prevalence. *Antimicrobial Resistance & Infection Control* **7**, 135 (2018).

13. He WY, *et al.* Characterization of an International High-Risk Escherichia coli ST410 Clone Coproducing NDM-5 and OXA-181 in a Food Market in China. *Microbiol Spectr* **11**, e0472722 (2023).

REVIEWERS' COMMENTS:

Reviewer #2 (Remarks to the Author):

The authors investigate the global transmission of CRE by characterizing the genetic diversity, antibiotic resistance, and virulence profiles of 7918 strains. Transmission of major sequence types are modelled across continents and potential transmission hubs are investigated. The authors have addressed my major concerns, including the details and biases that may arise from their methods. These methods are still evolving in its application to studying bacterial populations but this study is novel in the scale of it's application. I believe the statistical methods to be appropriate and valid for this work and I believe this study will help advance the field of phylodynamics.

Reviewer #3 (Remarks to the Author):

I have reviewed the revised manuscript and am generally satisfied with the authors' responses to my concerns. However, I would like to address the following minor points:

Terminology Change:

I suggest replacing "Conjugative mobilizable" with "mobile plasmids" throughout the manuscript. This is a minor linguistic adjustment that, I believe, would enhance clarity.

Revisions According to Q9 and Q.10:

Could you confirm whether the changes discussed in my concerns Q9 and Q.10 of the rebuttal have been incorporated into the revised manuscript? I would appreciate your verification on this matter.

Overall, the authors have effectively addressed all major concerns and commendations. Therefore, I recommend the manuscript be advanced for further processing.

1 **Response to Reviewer #2:**

2 **The authors investigate the global transmission of CRE by characterizing the**
3 **genetic diversity, antibiotic resistance, and virulence profiles of 7918 strains.**

4 **Transmission of major sequence types are modelled across continents and**
5 **potential transmission hubs are investigated.**

6 **The authors have addressed my major concerns, including the details and biases**
7 **that may arise from their methods. These methods are still evolving in its**
8 **application to studying bacterial populations but this study is novel in the scale of**
9 **it's application. I believe the statistical methods to be appropriate and valid for**
10 **this work and I believe this study will help advance the field of phylodynamics.**

11 **R:** We are honored by your positive comments. We thank you and the other reviewers
12 for your professional comments on the manuscript. The comments addressed biases in
13 various aspects of the study, such as its design, presentation of methodology, and
14 discussion. We recognize your diligence in addressing grammar in the manuscript, as it
15 is important to maintain the professionalism of the paper.

16 Carbapenem resistance poses a significant clinical threat and is expected to persist as a
17 long-term challenge in managing the spread of bacteria, including antibiotic-resistant
18 *E. coli*. Many methods for characterizing bacterial populations with respect to their
19 phenotypes are constantly evolving in epidemiology and bioinformatics. Improving the
20 management of antibiotic-resistant bacteria primarily involves implementing broader
21 and more current surveillance, consolidating and regularly updating large datasets, and
22 using innovative research techniques.

Response to Reviewer #3:

I have reviewed the revised manuscript and am generally satisfied with the authors' responses to my concerns. However, I would like to address the following minor points: Terminology Change:

Q1: I suggest replacing "Conjugative mobilizable" with "mobile plasmids" throughout the manuscript. This is a minor linguistic adjustment that, I believe, would enhance clarity.

R: Plasmids play a crucial role in facilitating horizontal gene transfer. This work discussed conjugative plasmids, mobilizable plasmids, and mobile plasmids. Plasmids can be classified into three types based on their sequence characteristics: conjugative, mobilizable, and non-mobilizable. Conjugative plasmids are capable of autonomous movement, whereas mobilizable plasmids can be mobilized through conjugation ¹. In this manuscript, both conjugative and mobilizable plasmids are referred to as mobile plasmids due to their inherent ability to transfer. We double checked all mobile plasmid-related presentations.

Revisions According to Q9 and Q.10:

Could you confirm whether the changes discussed in my concerns Q9 and Q.10 of the rebuttal have been incorporated into the revised manuscript? I would appreciate your verification on this matter.

R:We re-checked the answers for Q9 and Q10.

In Q9, we aim to show the proportion of CRECs within a specific phylogroup or in a particular locality harboring two or more CRGs. This was visually depicted by the purple area in Figure 1A, 1C, and 1D. The original text has been revised to include precise percentage values to prevent any potential misinterpretation.

The text has been revised as: "Phylogroup C had the highest proportion of strains with two or more CRGs, accounting for 8.76%. This was followed by phylogroup B1, which accounted for 3.40%. Africa had the highest proportion of strains containing multiple CRGs among all continents, with a percentage of 14.04%. Among individual countries, India had the highest percentage at 12.14%, followed by Israel at 8.66%."

Text revised: Lines 121 to 126 have been revised.

In Q10, we analyzed the percentage of sample sources (human, animal, and environmental sources) for different CRGs, as shown in Figure 1B. In addition, we conducted a comparative analysis of the proportion of specific CRGs across various sample sources. There are several explanations for this result. *Bla_{NDM}* exhibits the highest proportion of distinct resistance genes in human-derived CRECs. Additionally, *bla_{OXA-48}* exhibits a greater prevalence in human sources compared to environmental and animal sources. Given the considerable variation in the absolute number of strains across different sources, examining the relative proportion of identical resistance genes among various sample sources is intriguing.

The text has been revised as: The distribution of CRGs varies among strains from diverse sample sources. *Bla_{OXA-48}*-like was responsible for 30.87% (2208/7152) of CRECs isolated from human sources. The percentage was twice as high for animal-derived CRECs and six times as high for environmental-derived CRECs.

Text revised: Lines 134 to 137 have been revised.

Overall, the authors have effectively addressed all major concerns and commendations. Therefore, I recommend the manuscript be advanced for further processing.

R: We are honored by your positive comments. We appreciate your professional and careful corrections to the manuscript. We will make further revisions based on your suggestions.

Reference

1. Coluzzi C, Garcillán-Barcia MP, de la Cruz F, Rocha EPC. Evolution of Plasmid Mobility: Origin and Fate of Conjugative and Nonconjugative Plasmids. *Mol Biol Evol* **39**, (2022).